# Small AgNP in the Biopolymer Nanocomposite System

**DOI:** 10.3390/ijms21249388

**Published:** 2020-12-09

**Authors:** Małgorzata Zienkiewicz-Strzałka, Anna Deryło-Marczewska

**Affiliations:** Department of Physical Chemistry, Institute of Chemical Sciences, Maria Curie-Sklodowska University, 3, Sq. Maria Curie-Sklodowska, 20-031 Lublin, Poland; annad@hektor.umcs.lublin.pl

**Keywords:** nanocomposites, chitosan, biopolymer, small nanoparticles, in-situ reduction, silver

## Abstract

In this work, ultra-small and stable silver nanoparticles (AgNP) on chitosan biopolymer (BP/AgP) were prepared by in situ reduction of the diamminesilver(I) complex ([Ag(NH_3_)_2_]^+^) to create a biostatic membrane system. The small AgNP (3 nm) as a stable source of silver ions, their crystal form, and homogeneous distribution in the whole solid membrane were confirmed by transmission electron microscopy (TEM), scanning electron microscopy (SEM), and atomic force microscopy (AFM). The X-ray photoelectron spectroscopy (XPS) and Auger analysis were applied to investigate the elemental composition, concentration, and chemical state of surface atoms. It was found that ultra-small metallic nanoparticles might form a steady source of silver ions and enhance the biostatic properties of solid membranes. Ultra-small AgNP with disturbed electronic structure and plasmonic properties may generate interaction between amine groups of the biopolymer for improving the homogeneity of the nanometallic layer. In this work, the significant differences between the typical way (deposition of ex-situ-prepared AgNP) and the proposed in-situ synthesis approach were determined. The improved thermal stability (by thermogravimetry and differential scanning calorimetry (TG/DSC) analysis) for BP/AgP was observed and explained by the presence of the protective layer of a low-molecular silver phase. Finally, the antibacterial activity of the BP/AgP nanocomposite was tested using selected bacteria biofilms. The grafted membrane showed clear inhibition properties by destruction and multiple damages of bacteria cells. The possible mechanisms of biocidal activity were discussed, and the investigation of the AgNP influence on the bacteria body was illustrated by AFM measurements. The results obtained concluded that the biopolymer membrane properties were significantly improved by the integration with ultra-small Ag nanoparticles, which added value to its applications as a biostatic membrane system for filtration and separation issues.

## 1. Introduction

The nanoworld of the ultra-small structures is a rich source of new, functional materials and inspiration for science and solutions to everyday problems. Nanomaterials are an inescapable part of modern technology and the subject of heated debates and intensive research [1,2,3,4]. Their importance in this field cause formation of new products such as metallic nanoparticles, polymer particles, and polymer nanocomposites, silica-based materials, ceramics, and clays [5,6,7,8]. In this rich world, the nanocomposites based on metallic nanoparticles remain the most important. Nanocomposites suggest heterogeneous at the nanoscale and multiphase systems with significant properties of each component. In other words, the desirable combination of properties is obtained by the combination of materials with their own and unique properties. Generally, one or more phases with nanoscale dimensions (0-D, 1-D, and 2-D) such as noble metal nanoparticles are located on the surface or embedded in a matrix [9,10]. A tremendously important and topical issue is the new action taken in the selection of a suitable type of support or matrix. Biopolymers (natural polymers) are biodegradable, non-toxic, environment-friendly materials, equipped with unique surface properties, obtained from renewable resources, and can be successfully used as a nanocomposite matrix. The research and development of new types of nanocomposites is the result of the demand for materials better adapted to developed technologies. The composites based on biopolymers such as chitosan (CS) become important with particular attention paid to its biological properties [11] and applications in medicine [12], food industry [13], and agriculture [14,15,16,17,18]. These capabilities are directly related to the presence of amine functional groups [19]. For instance, amine groups can be present as the ionized forms NH_3_^+^ for effective electrostatic interactions with negatively charged bacterial cell walls. Due to the low-quality properties of biopolymers (mechanical properties and thermal stability), the nanocomposites attitude is the way for their improvement [20]. The improved biopolymers are widely used in various biomedical applications, such as tissue engineering, drug delivery systems, wound dressing, and antibacterial systems. Biopolymers can be modified by noble metal nanoparticles (gold [21], copper [22], and silver [23,24]), however, the fabrication of a fine and uniform composite have been widely used in the stabilization of microbial proliferation due to its anti-inflammatory effect and biostatic behavior [25,26,27]. A responsible approach to the synthesis of noble metal nanoparticles and nanomaterials like green chemistry concepts and enhance their stability is necessary for achieving their usability [28]. A very different concept of nanoparticle preparation refers to the direct involvement of the carrier surface in the synthesis of nanoparticles. What is more interesting, it is possible to use the functional groups building support structure as an at-place reducer. In before, in situ chemical reduction conception relied on the incorporation of the precursor on the surface through adsorption, deposition, or self-assembly mechanisms and their reduction in the second step by a dedicated reducing agent (formaldehyde, NaBH_4_, UV irradiation, etc.) [29,30,31,32]. Some disadvantages of this approach are the necessity to use reducing agents and the unsatisfactory effects related to the size of the resulting nanoparticles. For example, Taurozzi et al. [33] tried to synthesize AgNP in a membrane system by ex-situ and in situ reductions of the silver precursor. It was observed that there was no significant difference in cross-section morphologies of both cases. It was observed that some differences in the location of nanoparticles occur rather than having morphological and dimensional changes. However, in both cases the size of AgNP was significant (50–500 nm). The morphological changes toward size reduction are quite significant. Recent advances in synthetic methods and applications of silver nanostructures are searching for the possibility of building systems with small dimensions and stable behavior in the carrier environment. AgNP with small sizes can be obtained by reducing the process with sodium borohydride in the role of reduction agents. However, obtaining uniform particles is difficult by this approach. Some examples include the preparation of silver nanoparticles with a size of 5–10 nm by applying sodium borohydride and sodium citrate for preferential reduction, fast nucleation, and stable growth AgNP [34]. Silver particles that were 7–10 nm and had great uniformity in shape and size were prepared by thermal reduction of silver trifluoroacetate in isoamyl ether in the presence of oleic acid [35]. Well-defined, monodisperse silver nanoparticles with a diameter less than 10 nm were prepared also through silver nitrate reduction by aniline and dodecylbenzene sulfonic acid (DBSA) as reducing and stabilizing agents respectively [36]. Looking at scientific works on this subject, the vast majority of systems containing silver nanoparticles concern dimensions above 10 nm [37,38,39,40,41]. In this work, we propose the synthesis of biopolymer systems with silver nanoparticles with dimensions significantly below 10 nm using a non-complex procedure that does not require the use of reducing agents.

The system of silver nanoparticles deposited on a fibrous biopolymer substrate is important from the point of view of practical applications, especially biomedical ones. The most overriding elements include making the basis of wound repair systems with advanced and controlled drug release mechanisms [42,43,44]. When the AgNP is applied their stability and bonding strength may be responsible for various adhesion properties and provide a specified rate of Ag+ release. This in turn translates into antibacterial properties and determines the usefulness of such a system. Recent studies of similar multicomponent systems (chitosan/silver-NPs) systems suggest synergistic antibacterial effects achieved by combining chitosan with Ag-NPs, which increases the importance of water filtration systems [45]. The stability of the incorporated AgNP phase allows building improved nanofiltration systems for purification of drinking water and other substances [46,47,48]. The idea may be, e.g., a membrane system for the filtration of liquid systems (e.g., milk) with the simultaneous sorption of appropriate biomolecules, e.g., cholesterol [49,50,51,52].

The incompleteness of knowledge about biopolymer multiphase systems enforces undertaking research for enlargement and description of new properties and defining the relationships between particular elements of new composites. In this work, the CS nanofibers as an example of a biofunctional matrix decorated by AgNP were prepared and characterized. Regardless of the effectiveness in the antibacterial effect of silver nanoparticles deposited on the biopolymer surface in a reduced form (as in the work preceding the current research [41]), extensive efforts were made to strengthen and stabilize the biopolymer surface by trying to miniaturize metallic particles for creating a reservoir of silver ions, investigation of the nature of the interaction between AgNP and biopolymer surface, and building an improved homogeneity of the nanometallic layer on the surface of biopolymer fibers.

## 2. Results and Discussion

The chemical composition of surface layers of nanocomposites and functional materials intended for biostatic applications affects their properties, interactions with the surroundings, stability, and determines the safe and responsible usage and confirmed the properties assumed during their designing [52]. Therefore, determining information regarding the surface layer is crucial to find a better understanding of surface chemistry/properties relationships of investigated systems. Here, the XPS analysis to study the qualitative and quantitative surface composition and chemical state analysis of silver nanophase was applied. Moreover, the percentage of silver phase on the biopolymer surface as a mass and atomic percentages were estimated by this approach. Figure 1 shows the survey scan and high-resolution spectra of the BP and silver modified BP surfaces (BP/AgP and BP/AgNP). The details of the XPS analysis of investigated samples are presented also in Table 1 and Table 2. The XPS peaks of the main components were detected and labeled on the XPS spectra as O1s, C1s, N1s, and Ag3d and Ag3p (Ag3d and Ag3p signals were identified for BP/AgP and BP/AgNP signals). Mass concentration of silver increased from 10.4% for BP/AgNP to 13.3% for BP/AgP while the atomic percentages of silver increased from 1.4% to 2.7% for respective samples (Table 1). The increase in the silver content in the surface layer for the ionic way and their in-situ reduction suggests the greater probability of obtaining the homogeneous silver layer and confirms the better efficiency in incorporation the silver phase onto the biopolymer surface compared to the ex-situ procedure despite the similar amounts of introduced silver (The silver concentration in reference solution of AgNP prepared by reduction the same amount of [Ag(NH_3_)_2_]^+^ as for BP/AgP was determined as 43.5 mg/L by atomic absorption spectrometry). The [Ag(NH_3_)_2_]^+^ with a higher affinity to chitosan functional groups determines the higher enterability of silver to the biopolymer surface. The silver precursor in the form of the diamminesilver(I) complex resulted in a greater amount of ammonia as a stabilizer of the higher values of pH and exceeding the PZC value of chitosan (6.5–7) [53], which were responsible for the great adsorption of silver complexes.

Data specifying the concentration of surface-located silver, although important, should be supplemented with other valuable information such as analysis of the chemical state, degree of oxidation, and the number of individual chemical forms. The high-resolution XPS spectra of Ag photoelectrons (Figure 1B,C) revealed the presence of two peaks of a spine orbit doublet of Ag3d corresponding to 3/2 and 5/2 silver spin states. The position and heterogeneity of the photoelectron peak may be dependent on the size of nanoparticles. The general observation of the XPS signals show increasing the binding energy and decreasing the kinetic energy of the Ag Auger peak for BP/AgP sample revealing a reduction in the size of the nanoparticles. The widening of the Ag3d signals and shift of BE towards lower values may prove that the ionic state was maintained for this type of material. Detailed XPS data analysis is provided below. The positions of silver Ag3d5/2 and Ag3d3/2 peaks for BP/AgP samples were 367.4 eV and 373.4 and shifted by 0.7 eV towards lower values of the binding energy from the position typical for silver in the metallic state [54]. For BP/AgP sample the binding energy of Ag3d level doublet can be presented as two pairs of peaks. The first pair assigned as the Ag3d3/2A and Ag3d5/2A doublet at 373.9 eV and 367.9 eV, respectively, can be assigned as Ag–Ag bonding of silver metallic phase, but a slight shift of this peaks from the typical position (368.1 eV and 374.1 eV may indicate the current ionic state). The second pair of peaks assigned as Ag3d3/2B and Ag3d5/2B revealed the values of binding energy as 373.3 eV and 367.3 eV, respectively, and was shifted by 0.7 eV from the first pair suggesting a pronounced ionic state. Thus the binding energies below binding energies of the metallic state should be assigned to the Ag^+^ or Ag_2_^+^ states [55]. The XPS Ag 3d5/2 and Ag 3d3/2 core level binding energies for BP/AgNP appeared at 368.1 eV and 374.1 eV, respectively, which was in good agreement with bulk silver metallic values and means a fully crystalline form of silver for this sample.

Ex-situ prepared and deposited AgNP on biopolymer without additional signals or shift of binding energy indicating a lack of chemical interactions of silver atoms with other components (oxygen or nitrogen). For these samples, the presence of silver in the metallic state was confirmed previously by the XRD patterns. Thus, the stable ionic state of the silver in the BP/AgP sample was confirmed, which is important in terms of biostatic applicability. Ultra-small metallic nanoparticles (if stable) create a steady source of silver ions, which enables a functional biostatic system. Smaller particles with the greater surface area were compared with the same mass of larger particles because they released a greater amount of ions initially located on the surface. Thus, the smaller particles had a greater antibacterial effect related to the number of ions than the larger particles per unit mass. The Ag 3d Auger analyses of BP/AgP and BP/AgNP composites were carried out to further confirm the above results and especially to distinguish the states of Ag^0^ and Ag^+^. The kinetic energy of the Ag M_4_N_45_N_45_ and Ag M_5_N_45_N_45_ Auger peaks can be used to determine the chemical state of silver atoms [56]. For BP/AgP sample the Auger peaks Ag M_5_N_45_N_45_ and Ag M_4_N_45_N_45_ were located around KE 351.5 eV (1134.6 eV of BE) and KE 357.6 eV (1129.5 BE) respectively. The Auger parameters calculated for both peaks Ag M_5_N_45_N_45_ and Ag M_4_N_45_N_45_ equaled 719.4 eV and 724 eV. These significant shifts of the Auger parameter from position calculated for the BP/AgNP sample (presented below) to the low-energy region indicate the presence of the silver ionic state on the biopolymer surface. For comparison, the corresponding kinetic energies of Ag M_5_N_45_N_45_ and Ag M_4_N_45_N_45_ peaks and Auger parameters for BP/AgNP samples were 353.5 eV, 358.5 eV, 721.6 eV, and 726.6 eV, respectively. Both of them indicate the metallic silver phase and differentiate the data shown above for the BP/AgP sample. It should be emphasized that Auger analysis was particularly sensitive to changes in the chemical state of silver nanoparticles and allowed one to determine these properties even when the positions of the photoelectric peaks were very similar. Thus, it confirms the stable ionic state of ultra-small silver nanoparticles and indicates the success of this research.

The issue of the interaction between silver and the biopolymer is important due to the stability of silver nanoparticles and their potential release from the nanocomposite body. Very small silver nanoparticles with disturbed electronic structure and plasmonic properties may generate interaction between amine groups from a biopolymer body. Here also the stabilization of the [Ag(NH_3_)_2_]^+^ ions for in-situ reduction may take place. However the ionic state of silver atoms was confirmed for BP/AgP (and excluded for BP/AgNP) by Auger analysis and detailed analysis of Ag3d photoelectron signals (Figure 1 and Figure 2), the nature of the interaction between elements in the nanocomposite system and stabilization the [Ag(NH_3_)_2_]^+^ complex by the biopolymer surface was clarified by analysis the chemical state of the adjacent atoms. The high-resolution XPS spectra of N1s (Figure 3A) revealed the presence of single spin states for N1s with shifts of binding energy within the main signal and several partial peaks. For the initial chitosan sample (Figure 3A) the amine groups (–NH_2_ groups and N1sA signal) might be protonated (due to the acidic condition of their preparation) and thus acquire a positive charge (–NH_3_^+^) that is visible as a shift of BE to higher values (signal marked as N1sB) [57]. The quantitative XPS analysis suggests the presence of –NH_2_ groups predominantly (94%) and a slight but noticeable amount of the protonated form (6%) in the pristine form of the support. The high-resolution XPS spectrum for the N1s core level for the BP/AgP (Figure 3B) sample revealed the same chemical state of nitrogen atoms (N1sB and N1sC) and additional signal (N1sA) in 398.5 eV. The presence of the silver ammoniacal complexes might cause decreasing the N1sC signal due to the higher pH value of the silver precursor solution. This could be the reason for lowering the N1sC signal and the factor responsible for promoting the cation adsorption. This type of bond can be assigned as an Ag–N connection and corresponds to the nitrogen–silver interaction in the [Ag(NH_3_)_2_]^+^ complex and nitrogen–silver interactions in NH_2_–silver (amine groups from biopolymer) [58]. At this point, neither of these cases should be excluded. However, the presence signal at 398.5 eV with a significant percentage (10%) may indicate the presence of the stabilized silver complex and the important role of the biopolymer surface in maintaining the essential properties of the metallic precursor by guiding its gradual reduction and building a homogeneous metallic layer. The AgNP in the BP/AgNP sample does not show such properties. In this case, the functionality of the surface visible by photoelectron spectroscopy is limited to the main signal marked as N1sA related to –NH_2_ groups.

Figure 4 shows the C1s photoemission bands observed of all investigated samples. The C1s core level was found in the range of 282–290 eV. The deconvolution of the experimental C1s spectrum for chitosan component suggests the presence of three general peaks: C1s A core level at 287.6 eV of the C–C aliphatic group (Figure 4A), C1s B core level at 286.1 eV of C–O or C–N groups as strong carbon and nitrogen/oxygen bonds (Figure 4B) and C1s C peak at 284.7 eV corresponding to the acetyl groups from chitosan backbone (O–C–O; Figure 4C). The obtained values of C1s components correctly matched to literature data [59]. The difference compared to the modified samples concerns the intensity of individual components due to the presence of a stabilizer: poly(ethylene oxide) (PEO) and poly(propylene oxide) (PPO). As known, PEO was used as also an auxiliary material for the BP electrospinning, and any eventual interaction with silver atoms and the connection mode between the compositions –OH and –O^−^ bonds in the PEO chains and modifiers should be tested. Here, from the carbon C1s bands no unusual signal shifts were observed. An increase in the signal associated with the stabilizer aliphatic chain C1s A was observed especially for the BP/AgNP sample (change from 14.5% for BP sample to 47.3% for BP/AgNP). For the O1s photoelectron line, the results of BP and BP/AgP and BP/AgNP were summarized in Figure 4D–F. Two peaks were identified at 531.1 eV as C=O chemical bonds in *N*–acetylated–glucosamine units and at 532.6 eV according to C–O–H groups [60]. The peaks at 532.5 eV were assigned to C–O–H, while the peak at 532.9 eV was assigned to the O–C–O chemical bonds. The signal analysis shows the presence of bonds typical of a biopolymer and a change in their intensity due to the presence of the stabilizing polymer. At this point, it is difficult to specify the potential interactions of O1s and oxygen atoms with silver nanoparticles.

The XRD pattern of biopolymer support (chitosan nanofibers) and biopolymer modified by AgNP nanocomposites exhibited characteristic crystalline peaks corresponding to the (020), (110), and (201) lattice planes of the crystalline P2_1_2_1_2_1_ lattice at 2θ = 10 and 2θ = 20 degrees and interplane distances of 8.84 Å and 4.44 Å correspondingly (Figure 5A). The intensity and separation of individual peaks were related to the changes in the degree of crystallinity of various samples. The calculated degree of crystallinity for BP was about 46%, whereas for BP/AgP and BP/AgNP these values were slightly higher (68.5% and 53.4%for BP/AgP and BP/AgNP respectively presented as an inset in Figure 5A). Since all samples had equal initial degrees of crystallinity (the same types of the support—BP) one could conclude that the chitosan fibers might be exposed to factors limiting the amorphous structure during their modification by the silver phase, both in its ionic and metallic form. The greater the pH and the presence of additional amine groups of the silver diammine complex could impact the self-assembly behavior of biopolymer chains to may be generate the additional stabilization and reduction of the amorphous structure by hydrogen bond formation between amino and hydroxyl groups and regularity in the biopolymer structure. The size of Ag particles from the perspective of the size of the crystallinity region was also evaluated by X-ray diffraction. The XRD analysis was applied for describing the bulk structural information in the form of the mean value of crystallite size for the whole nanocomposite sample The wide-angle XRD (Figure 5B) patterns recorded for nanocomposites with the silver phase (BP/AgP and BP/AgNP) revealed the presence of intense diffraction signals at 38.1°, 44.3°, 64.5°, and 77.5°, according to Ag(111), Ag(200), Ag(220), and Ag(311) lattice planes of the silver cubic structure (Joint Committee on Powder Diffraction Standards—JCPDS, File No. 04-0783). The full width at half maximum (HWHM) values of Ag peaks was used to calculate the average size of the silver crystallites according to all visible lattices (insets in Figure 5B) [61,62]. The calculations suggested that the silver particle size taking into account their crystallinity area was 20 nm, 22.3 nm, 25 nm, and 21 nm for respectively lattice planes (111, 002, 022, and 311). The size of the silver phase (size of crystallites) for the BP/AgP sample was significantly lower and equals to 3.2 nm, 2.8 nm, 2.5 nm, and 3.6 nm for the same lattice planes. Here, a decreasing intensity and widening of the half-width of peaks was observed according to the nano effect, which was related to the broadening and reduction of the pattern from nano-sized objects.

The experimental small-angle X-ray scattering (SAXS) curve contains unique information about the structure of the heterogeneous nanomaterials. In general, SAXS can be used for the investigation of the micro- or nanoscale structures by determining average particle size, shape, and their distributions in the matrix with different electron densities. The one requirement is the occurrence of specific structural heterogeneities (electron density differences) between scattering particles and the background phase. SAXS effect is observed due to the presence of nanometric structures (particles) in the investigated samples [63]. When the differences in electron densities between specific areas in the sample are significant, the significant changes in the SAXS spectra are noticeable. Due to the high sensitivity to the size, morphology, and interfacial properties of nanomaterials, SAXS can be applied to analyze the structural changes in the biopolymer matrix after modification by the metallic silver phase. Investigated systems include the crystalline, amorphous, and background regions with characteristic electron densities and their structural fingerprint is illustrated by the experimental SAXS curves (Figure 6).

Figure 6A shows the experimental SAXS data plotted as a function of the length of the scattering vector q (scattering intensity vs. length of the scattering vector) for all investigated systems. It was noted that the intensity and general course of SAXS curves vary for individual samples suggesting various microstructure and X-ray scattering abilities. For a better visibility, mutual differences in the initial range of the curves were presented in a logarithmic scale (Figure 6B). The intensity of the scattering signal in the initial range of the SAXS curve (q < 0.05 Å^−1^) was the strongest for the biopolymer modified by silver ions and in-situ reduced to the metallic state (BP/AgP). The lowest intensity for material without the silver nanophase (BP) was observed. It was the first evidence that significant changes in the nanocomposite structure occurred after their modification by the silver phase and these changes depended on the form of silver precursor and their amount in biopolymer body. A high level of scattering intensities reveals a single correlation with the presence of a significant amount of the silver nanophase and improved scattering ability. The initial range of the SAXS curve q < 0.04 Å^−1^) for BP/AgN exhibited a smooth and stable course suggesting a higher amount of uniform scatterers than for the BP/AgNP sample. The lack of nanoparticles on the biopolymer surface (BP sample) causes the flat scattering profile without any inhomogeneous area and points on the scattering curve. It is worth mentioning that the SAXS effect and changes of the SAXS curve intensity are straight proportional to the square of the difference of electron density of the particle and the environment. So, the SAXS effect can appear regardless of whether the electronic density of the scattering systems is lower or higher than the electron density of the environment. In the simplest terms, it can be expected that the observed effect will be the sum of scattering by metallic particles and pores in the biopolymer body. To quantify these changes, the values of the sizes of electron density inhomogeneity described as *d* = 2π/q where q = 2 πsinθ/λ were calculated for peaks visible on the SAXS curves. Analyzing the course of the curve, for the BP sample, in the initial range of q values, it was observed that experimental curves revealed the presence of only one peak at the position of q = 0.04 Å. In this case, the scattering effect from pore systems, or other characteristic systems of the biopolymer material itself should be expected. On the other hand, if the dispersion system with this specific electron density is a feature of the biopolymer, a similar peak should be noticeable for all materials regardless of the modification. Additionally, indeed such an effect has been confirmed here. The size of the electron density inhomogeneity *d* calculated for the first peaks (q = 0.04 Å^−1^) equals around 150 Å. Comparing the scattering capacity of the sample modified with silver nanoparticles in a reduced form (BP/AgNP), the increasing scattering capacity was detected according to the elementary SAXS principle that nano-sized objects are the centers responsible for the scattering effect. Secondly, an additional peak at the position of q = 0.03 Å^−1^ was observed. The presence of the heterogeneity within the lower values of the q vector indicates a larger size of particles. The size of the electron density inhomogeneity d generated in BP/AgNP for a peak at q = 0.03 Å^−1^ was noticeable and equaled around 200 Å. This may suggest the size of the silver nanoparticle layer below 20 nm. This was consistent with the properties of the AgNP solution used for biopolymer modification. The AgNP solution exhibited well-defined optical properties indicated by the presence of an intense absorption band at 420 nm caused by the oscillation of all the free electrons in the particles (Figure 16 in experimental part). Both, the position and shape of the plasmonic signal indicate the presence of spherical particles with a diameter of 20 nm. The situation looks more interesting for the BP/AgP sample. In this case, the signal for a biopolymer body (q = 0.04 Å^−1^) was observed, however, their intensity was significantly lower and the half-width was widened. This points to the existence of other explanatory factors for the scattering effect. The peak associated with silver nanoparticles with a size of 20 nm (q = 0.03) was practically invisible. This implies the lack of nanoparticles with a size from 10 to 20 nm. Similarly, in terms of higher q-values, i.e., smaller objects, no cystic points were observed. Moreover, it should be noted that the silver phase in the form of silver ions (AgP solution) does not generate the SAXS effect. However, the scattering capacity of this sample had the highest level. This may suggest a significant reduction size of the crystalline metallic phase, which was responsible for increasing the scattering effect. Analysis of the size of silver nanoparticles in the BP/AgP sample was discussed in greater detail below. At this point, the presence of small size nanoparticles was signaled by SAXS. In the case of the BP/AgP sample, the in situ formation of colloidal nanoparticles by functional groups on the chitosan body was accompanied by color changes of a silver precursor in a short time, from completely colorless and transparent to dark brown observed only on a biopolymer clipping, which is typical for small silver nanoparticles. Whereas, the solution remained colorless. The lack of any color of the precursor solutions suggests a lack of formation of metal nanoparticles in the solution phase. As per the above, it can be stated the BP/AgP sample contains very small silver nanoparticles, which were obtained using the reducing properties of the functional groups on the surface of the biopolymer support. After dimensioning the metallic phase by other techniques (previously by XRD and below by SEM and AFM) this result was fully justified. Morphological changes of the biopolymer material and dimensioning of the metallic phase on the biopolymer surface in the case of the BP/AgP sample suggest attaining the goals and preparation ultra-small silver nanoparticles on a biopolymer surface by in-situ reduction of the silver precursor. Polysaccharides contain hydroxyl and amino functional groups and can serve both as effective metal-reducing and capping behavior for in-situ reduced nanoparticles. The presence of amino groups directly affects this ability to interact with other molecules. It is, therefore, appropriate to consider the mechanism of reduction and stabilization of the silver ions by the biopolymer surface (Figure 7). In the chitosan structure, three types of reactive functional groups: a primary amino group and both primary and secondary hydroxyl groups were embedded (Figure 7). Among them, amine groups on chitosan structure can play a key role in the reduction of silver ions to the silver metallic state. The important feature is also their random location in the biopolymer chain and the ability to create hydrogen bonds. In-situ reduction of silver ions on the chitosan biopolymer and local stabilization are crucial elements in guaranteeing uniformly distributed, ultra-small, and monodispersed particles. This is directly reflected in creating coordination bonds between chitosan and metal ions and its reduction abilities. During reduction, the coordination between nitrogen from primary amine groups on biopolymer and silver ions may be responsible for decreasing the potential EAg^+^/Ag^0^ and facilitating the reduction process. The new (as-prepared) silver nanoparticles can be stabilized at-place, by coordination of the silver atoms into several amino groups from the chitosan body just during their reduction. Frattini et al. [64] suggest also that the number of amine groups in the capping or reducing agent has a strong effect on the size of the resulting metallic nanoparticles. Here, the type of silver precursor (in the form of diamminesilver(I) complex) is significant. The diamminesilver(I) complex contains silver ions coordinated by two NH_3_ groups. It is also a metallic precursor containing volatile components (ammonia) favoring the reduction of silver ions into nanoparticles under relatively gentle conditions [65,66]. In this case, the coordination of silver ions by amine groups from chitosan further reduces the silver potential E_Ag_^+^/_Ag_^0^.

From the small-angle scattering data, the size distribution of the scattering particles can be determined to assume their shape. The volume-weighted particle size distribution (Dv(R)) function illustrates the radii R of the particles that are present in the sample. The height of the Dv(R) function was proportional to the volume of particles that can be found within a given size interval. A prerequisite for determining the correct function of particle size distributions should cover both Guinier [67] and Porod [68] ranges (q < 0.1 Å). The experimental SAXS data were treated according to the SAXS theory for Dv(R) determination. The obtained calculations were presented in Figure 8. The BP sample as unmodified biopolymer support (Figure 8A) contained scattering inhomogeneities of relatively large dimensions of 30 nm. To determine their exact shape, detailed SAXS analysis and specific calculations should be made taking into account particle systems with longitudinal, flat, or spherical morphology. These data were not included in this work. Here, only the spherical systems of the introduced metallic phase were focused. AFM analysis of the BP sample revealed that the surfaces of the biopolymer nanofibers without silver nanoparticles were smooth, continuous, and well developed. No significant surface defects, pores, or concrescences were visible (Figure 8A).

The algorithm used in easySAXS was based on homogeneous, spherical particles where interparticle interaction and agglomeration effects were at least not too pronounced. Figure 8B shows that BP/AgNP system was relatively not very homogeneous. The Dv(R) function shows one well-defined peak with a maximum of 15 nm. However, the peak characteristics (quite broad and tailing) suggest that silver nanoparticles prepared by chemical reduction before deposition on the biopolymer material show limited uniformity in size. Although the system stability was satisfactory, the formation of nanoparticles of various sizes was not avoided. In this case, the average radius of particles was calculated as 15 nm. The suitable AFM image suggests, that the homogeneous fiber surface changed significantly after the incorporation of the metallic nanoparticles, which existed as small, mostly spherical, and semi-spherical objects on the surface of the nanofibers (Figure 8B). The silver nanoobjects were randomly located on the surface, without noticeable signs of agglomeration. The size of the AgNPs, determined from AFM images, was 20–50 nm and even greater aggregates. The sample BP/AgP contains metallic nanoparticles with a much higher degree of homogeneity (Figure 8C). The Dv(R) function contains one well-defined peak, indicating a scattering particle size of about 2–5 nm. More detailed analysis results relate to: the most frequent radius: 3 nm, average radius: 4.9 nm, and R20, R50, and R80 parameters (where the cumulative undersize was 20%, 50%, and 80% respectively) as 2.64 nm, 4.83 nm, and 7.31 nm respectively. The size of silver nanoparticles obtained by scattering techniques was confirmed also by X-ray diffraction (Figure 5) and directly correlated with AFM images of investigated samples (Figure 9) and further by TEM and SEM images (Figure 10 and Figure 11). The surface of the biopolymer material in the BP/AgP sample examined by AFM was completely different. Here ultra-small nanoparticles of the silver phase were located on the surface and covered almost all external surfaces of the fibers with a high degree of homogeneity. The silver ion reduction mechanism based on -NH_2_ groups evenly distributed on the surface of the biopolymer allowed for an even location of metallic particles. The use of amine groups as a reductant also enabled the location of silver nanoparticles on the inner layers of the material, which were accessible to silver ions, were inaccessible to their reduced forms, and might not be modified during the deposition of previously prepared silver nanoparticles. Such a high degree of homogeneity in the distribution of nanoparticles cannot be achieved by other methods. AFM topography as 3D projections (Figure 9) of the BP/AgP nanocomposite allowed us to accurately characterize the roughness of the solid surface and assess the degree of their coverage after the formation of the silver nanoparticles. AFM images of a wider range of material (Figure 9c) suggest the full filling of the biopolymer surface and confirmed the high quality of the modifying layer.

The morphological properties of the AgNP and their small sizes were evaluated by high-resolution transmission electron microscopy (Figure 10). The TEM micrographs revealed that the nanoparticles released from the system (Figure 10A) were well-developed crystalline systems with dimensions of about 5 nm. The amount of silver nanoparticles released as a result of ethanol extraction was relatively small. Among them, there were no objects of significantly smaller sizes or agglomerated forms. All this makes the silver nanoparticles connected with the biopolymer surface remain stable, and their susceptibility to possible release into the environment can be limited. Silver nanoparticles with dimensions of 2–5 nm were visible on a small fragment of biopolymer fiber that was extracted from the system (Figure 10B). The AgNP arrangement on the solid surface suggests a relatively homogenous distribution. Moreover, blurred boundaries between the metallic and biopolymer phases were observed as biopolymer regions that surround the AgNP. This was especially visible in the enlarged areas of the TEM photo. Finally, even such small nanoparticles were crystallographically well-formed (Figure 10C). The crystal structure was confirmed by visualized lattice planes (Ag111) according to the silver phase crystallography. The morphology of the nanoparticles deposited on the biopolymer surface and their dispersion is important in the development of new and functional nanocomposites. Especially when the functional membrane systems were considered, the dispersion of the potential active phase was significant. In this work, the preliminary information about the dispersion of the metallic phase obtained by XRD and SAXS techniques was confirmed with the electron microscopic data (SEM; Figure 11). SEM images show similar particle shapes, their narrow size distributions, and the real three-dimensional morphology of the synthesized nanocomposites (Figure 11A–D). Moreover, the effect of ultra-small silver nanoparticles was confirmed for BP/AgP sample. SEM imaging shows that the entire surface of the biopolymer fiber was available, firstly for the precursor and secondly for silver in the nanoparticles form. In the case of the BP/AgNP sample (Figure 11E,F), it was observed the preferred orientation of nanoparticles on the surface of the biopolymer fiber and practically their absence in the deeper layers of the membrane. In this case, SEM images provide valuable information regarding the high purity of nanocomposites and the low degree of particle aggregation. This lack of agglomeration for the BP/AgP sample was confirmed by the result of SEM/EDX in the form of maps of AgL spectra lines (Figure 11A) and correlated with a similar comparison for the BP/AgNP sample where such a phenomenon unluckily was observed (Figure 11F).

However the size of nanoparticles is important, the surface topology of nanoparticle-solid surfaces (nanoparticles integrated with the surface) and especially spatial distribution and nanoscale features were significant due to the impact of the stability of biopolymer-AgNP systems. It turns out that the problem of stabilization has two meanings for both ingredients. Firstly the immobilization of the silver ions on the biopolymer surface and their in-situ reduction to the metallic state improves the properties of the chitosan carrier by an improvement of its thermal resistance (TG/DSC thermal analysis). Secondly, it enables one to obtain and maintain ultra-small silver nanoparticles in the functional solid matrix. It should be noted that silver ions in the form of [Ag(NH_3_)_2_]^+^ ions are highly unstable and maintaining such small silver nanoparticles even in the liquid phase is a real challenge. Considering the first case, the thermal analysis of the tested systems was carried out and the results are presented in Figure 13.

To study the thermal stability of the investigated samples, the thermal analysis in the synthetic air atmosphere was carried out and experimental TG, DTG, and DSC curves were obtained from room temperature to 950 °C. The TG/DTG-DSC curves of BP and silver modified samples (BP/AgP and BP/AgNP) are shown in Figure 12. In general, the thermal stability of the BP/AgP and BP/AgNP were very similar (Figure 12A,B). The similarity concerns the characteristics of the thermal decomposition illustrated by the characteristic signals of DTG and DSC. Based on thermal curves, three main stages of thermal degradation could be distinguished.

The first mass loss occurred in the range of 30–100 °C and was related to the dehydration process where the endothermic peak in the DSC curve appeared at 51 °C, 57.6 °C, and 76 °C for BP/AgNP and BP/AgP samples respectively. The water content was not significant and equaled 2.5%, 6.5%, and 6.1% for the above-mentioned samples. All samples showed thermal stability above approximately 200 °C when the decomposition started as the onset of weight loss and an increase in the thermal effect (exo effect on DCS curves).

The decomposition point was higher for BP/AgNP (210 °C) and the highest for the BP/AgP sample (250 °C), which means the improved thermal stability after silver modification. The first step of decomposition occurred in two ranges: 200–350 and 350–500 °C to BP, 210–400 and 400–550 °C to BP/AgNP, and 250–450 and 450–600 °C to BP/AgP and can be assigned as oxidation steps of organic matter. Some differences between BP and silver modified BP samples were related to the presence of PEO molecules in the system in the role of silver nanoparticles stabilizer. In this case, the signal at 400 °C (for BP) and 486 °C and 512 °C for BP/AgP can be related to the PEO presence in the samples, which is involved in building the silver phase on the biopolymer surface. The final residues of BP (7.5%) and BP/AgNP (9.2%) and BP/AgP (14%) showed silver content in polymer systems. From the obtained data it can be concluded that the silver content was the highest for the sample synthesized by the ionic route (BP/AgP sample). The improvement of thermal stability in the case of the BP/AgP material compared with BP/AgNP was related to the presence of a low-molecular silver phase on the surface of the biopolymer fibers, which created a protective layer and improved the thermal properties of the carrier.

The functionality of membranes modified with silver nanoparticles and the effect of their small dimensions were investigated by determining the antibacterial properties. In that case, the interactions between the very small particles (below 5 nm) and the chitosan nanofibers confirmed by XPS may suggest the improved antibacterial effect. Smaller particles had a greater surface area than the same mass of larger particles. Since the release of ions comes from the surface, the greater the surface area, the greater the release of ions. Presumably, the reason why the smaller particles had a greater antibacterial effect is that they release more ions than the larger particles per unit mass. These suggestions were confirmed by an experimental test of the biostatic properties of the BP/AgP sample and previously investigated antibacterial properties of BP and BP/AgNP samples [41].

A zone of the inhibition test (Kirby–Bauer Test) was applied to measure BP/AgP resistance to bacteria growth (Figure 13). The BP/AgP nanocomposite showed an 11–12 mm inhibition zone for the *Escherichia coli* strain and 14–15 mm inhibition zone for the *Staphylococcus aureus* strain, suggesting a higher biostatic activity than composites modified by AgNP (10 mm and 13–14 mm inhibition zone for the *E. coli* and *S. aureus* respectively). They demonstrated potential antimicrobial properties, highly influenced by the nanoparticle dimensions. The results show that the small size of the nanoparticles and their homogeneous distribution on the support surface caused greater biocide activity than BP/AgNP material with the greater size of AgNP. Thus the effect of increasing the bioactive properties with decreasing size was confirmed by increasing the inhibition zone from 10 and 13 (for BP/AgNP) to 11–12 and 14–15 mm (for BP/AgP) for *E. coli* and *S. aureus* respectively.

Moreover, neither strain showed a significant zone of inhibited growth around BP alone discs. Furthermore, it was found that AgNP suspension inhibited the growth of bacteria in a dose-dependent manner. When the same amount of AgNP (1 µL of AgNP dropped on the composite discs and 1 µL used directly) was used, the better activity of AgNP stabilized on the biopolymer was observed (the synergy between biopolymer and silver nanoparticles). The detailed biostatic activity of BP alone, AgNPs alone, and BP/AgNPs are presented in our previous work [41] and should be compared with a significantly more active BP/AgP system.

Atomic force microscopy (AFM) can be used to evaluate the surface morphology of a biological specimen changed as a result of interaction with a harmful factor. Figure 14A shows an AFM image of *E. coli* control cells. The cell exhibited a characteristic rod shape with a size above 5 µm. The *E. coli* cells before treatment were characterized by a homogeneous surface without visible and significant damage. Figure 14C–F illustrates *E. coli* bacteria after exposure to the BP/AgP nanocomposite. After contact of *E. coli* with BP/AgP, some features of bacteria destruction were defined. Mild and extensive membrane degradation effects, additional debris, and small fragments located on the bacteria surface and their vicinity and atypically local accumulation of cellular fragments were observed (Figure 14C). Moreover, significant cellular collapse, lysis of the cell contents (Figure 14E) and holes, and shortages of biological material were visible (Figure 14F). The cross-section profiles were created by selecting one line along with the bacteria cell of a two-dimensional AFM scan (presented as an inset in Figure 14B,D) and subsequently plotting as distance vs. height data. The roughness of the bacteria body became different after AgNP treatment. Here the Ra, Rq, and Rmax were higher for damaged bacteria and clearly illustrated the structure and nanoarchitecture features as the reflection of the interactions of a material with its external environment. Furthermore, cross-section line profiles were generated for untreated bacteria and bacteria after contact with the BP/AgNP surface and presented in Figure 14B,D, respectively The cross-section of bacteria without damage was smooth and free of unexpected surface irregularities, while such features were observed for the sample after exposure to AgNP. The set of morphologies observed indicates cell damage and confirmed the presence of steps mechanism to the cytotoxic effect of small AgNP against *E. coli* as an example of real microbial biosystems. The explanation of the visible changes is discussed below and illustrated in Figure 15.

Different antibacterial actions may have been involved in creating a coordinated bioactive strategy of silver-biopolymer materials. The basis of the mechanisms is the ability of a continuous release of silver ions by the surface of nanoparticles [69]. The probable mechanism of interaction of the AgNP with bacteria cells can be found in thematic scientific works [70,71] and presented schematically in Figure 15. The general (not all) interactions of nanosilver with the microbial cell may concern:When the Ag^+^ species are present in the system, disruption of a cell wall membrane and cytoplasmic membrane may be performed by adhesion the Ag ions to the cell wall due to electrostatic attraction and affinity to sulfur proteins. As an effect, the pores, weakness, and defects are generated, which may consequently lead to perforation of the wall and membrane and opens the entrance to the center of the cell (excerpt 1 in Figure 15).The membrane denaturation can be also induced causing the incorrect transmembrane ATP generation (excerpt 2 in Figure 15).The disintegration of the covered elements through denaturation and pouring out of the cell contents into the environment (lysis) [72].Then the penetration of the cell by silver species becomes possible. When the AgNP and/or Ag ions are present in the bacteria body and may interact with cellular structures. For instance denaturation of ribosomes may occur resulting in inhibition of protein synthesis. Similarly, the cell enzymes can be deactivated resulting in production of reactive oxygen (ROS; excerpt 3 in Figure 15).Next to this, the interaction with nucleoid may generate DNA damage [73,74] by impairing their replication according to the potential binding of silver and ROS with deoxyribonucleic acid (excerpt 3 in Figure 15).

What is more, especially, the small AgNP plays an important role in this issue. Ultra-small AgNP has facilitated the ability to penetrate bacterial cell walls due to their nanoscale size. The sieve-like effect may result in faster import of ultra-small silver nanoparticles into the cell interior. As was mentioned, smaller particles had greater surface area than the same mass of larger particles, which was directly related to the greater number of silver ions than the larger counterparts per unit mass. In this case, ultra-small silver nanoparticles on the surface of the biopolymer, which were also highly stable, constituted an excellent reservoir of silver ions that can attack microorganisms. The reason for the activity may be related to the faster penetration of silver factors related to both the improved ability to produce silver ions by small particles and the size of the nanoparticles themselves. Higher content of silver ions and a more durable system of their release means also greater effectiveness of interaction with atoms of the biological system (sulfur atoms). As was shown above, the small size of the nanoparticles and their homogeneous distribution on the support surface cause greater biocide activity than BP/AgNP material containing a significantly greater size of AgNP.

## 3. Materials and Methods

### 3.1. Reagents and Materials

The CS from crab shells (Poly-(1,4-β-D-glucopyranosamine, MW 110,000 with the degree of deacetylation 0.74) was purchased from Sigma-Aldrich (Munich, Germany). Polyethylene oxide (PEO), silver nitrate (≥99.0%), poloxamer (P123 with a molecular weight of around 5800 g/mol and nominal chemical formula: HO(CH_2_CH_2_O)_20_(CH_2_CH(CH_3_)O)_70_(CH_2_CH_2_O)_20_H, was used as a stabilizer of AgP and AgNP), sodium hydroxide, and ammonium hydroxide (28.0–30.0% NH_3_ basis) were purchased from Sigma-Aldrich. Aqueous acetic acid (≥99.7%) was purchased from POCH (Avantor Performance Materials Poland S.A, Gliwice, Poland). The biopolymer scaffold was prepared from a biopolymer solution according to the procedure presented in a previous paper [41]. All solutions were prepared with MilliQ standard water. All reagents were used without any further purification.

### 3.2. Preparation of the Colloidal Silver Solution

The diamminesilver(I) complex [Ag(NH_3_)_2_]^+^ as AgP was prepared by the addition of NaOH (concentration 5%) to AgNO_3_ (0.3 M). The silver(I) oxide in the form of brown solid was immediately produced according to the precipitation mechanism. Of freshly precipitated silver(I) oxide 1 g was separated from the solution and dissolved in a stoichiometric amount of concentrated ammonium hydroxide to obtain the diamminesilver(I) complex. The obtained solution was designated as AgP. For the synthesis of AgNP, the diamminesilver(I) complex can be reduced by formaldehyde in the presence of a P123 symmetric triblock copolymer (For this purpose, 5 g of P123 was completely dissolved in 50 mL H_2_O. Next, the specified amount of the diamminesilver(I) complex was carefully added to this solution. The reduction of AgP to AgNP occurred through the introduction of the formaldehyde (at least several droplets) to the above-mentioned solution.

### 3.3. Preparation of the Biopolymer/Silver Composites and Their Bioactivity

The biopolymer scaffold (chitosan nanofibers) had the form of an elastic and thin solid sheet whose undoubted advantage is it is easy to form and further process. Small pieces of the solids (1 cm^2^) were immersed in 25 mL of AgP (50 mL H_2_O + the same amount of [Ag(NH_3_)_2_]^+^ as for AgNP) and 25 mL of freshly prepared colloidal silver solution (AgNP). The solids were immersed for 24 h and treated with gentle mixing in an incubator at room temperature. The samples were gently separated from the solutions and air-dried at room temperature. Biopolymer/silver nanocomposites were used for structural, textural, and morphological characterization. The materials considered in this work were designated as BP (chitosan nanofibers as biopolymer scaffold without the silver phase), BP/AgP (biopolymer/silver nanocomposite obtained by impregnation the biopolymer scaffold by silver precursor), and BP/AgNP (biopolymer/silver nanocomposite obtained by deposition of the AgNP on the biopolymer scaffold). The spectroscopic characterization of the AgNP solution and the steps of BP/AgP sample preparation are shown in Figure 16. It should be noted that the in-situ reduction of the silver ions from the precursor solution was observed by changing the color of the biopolymer body just during impregnation (from the white color typical for the biopolymer to brown typical for the silver nanophase).

The antibacterial activity of the BP/AgP sample was investigated by a zone inhibition method. *Staphylococcus aureus* ATCC 25923 and *Escherichia coli* (ATCC 25922) were used as indicator strains. Petri dishes with MH agar were inoculated with 100 µL of bacterial cultures (suspensions containing around 1 × 10^8^ colony forming units/mL). The BP/AgP solid was cut into 6 mm diameter discs. Three pieces per plate were placed onto the dishes containing bacteria and incubated for 24 h at 37 °C (*Escherichia coli*) and 32 °C (*Staphylococcus aureus*), respectively. The antibacterial potency was determined by measuring the diameter of the inhibition growth zone around each disk or well. The results of the antibacterial activity of BP/AgNP and BP samples and details of the antibacterial evaluation procedure were presented in our previous work [41].

### 3.4. Measurements and Calculations

The obtained biopolymer/silver nanocomposites were evaluated by powder X-ray diffraction (XRD) using the Empyrean diffractometer (PANalytical, 2013, Malvern, UK) equipped with CuKα radiation (λ = 1.5418 Å) in the range of diffraction angles 10–70 degrees of 2θ. The X-ray diffraction analysis system was set to 40 kV and 30 mA (during analysis) and equipped with a PIXcel3D detector used in a scanning line 1D detector mode. The small angle X-ray scattering (SAXS) analysis was performed using the same device by applying transmission geometry in the range of 0.1 to 4 degrees of 2θ. Experimental data were treated by EasySAXS (PANalytical, Malvern, UK) software as a toolbox for SAXS data analysis. The Ag crystallite size L_Ag_ was determined by the Scherrer equation from the full width at half maximum of the X-ray diffraction peaks [75]:L_Ag_ = kλ/βcosθ(1)
where k is a dimensionless shape factor defined as constant related to crystallite shape (k = 0.9), λ is the X-ray wavelength (1.5418 Å), θ is the peak position in radians, and β is the full width at half maximum of the peaks located at 2θ position. The fitting of the X-ray patterns was performed using WAXSFIT software [76].

Auger analysis and X-ray photoelectron spectroscopy (XPS) allows analyzing the surface elemental composition, chemical state information by the binding energy of all components are useful for confirmed possible interactions of individual atoms between themselves and describing their form on the solid surface within nanometers. X-ray photoelectron spectroscopy data were collected on the Multi-chamber UHV System, Prevac (2009, Rogów, Poland) using the hemispherical analyzer Scienta R4000 by monochromatic Al Karadiation from high-intensity source MX-650, Scienta (Uppsala, Sweden). All of the binding energies were calibrated to the C1s peak. High-resolution XPS data were obtained for Ag3d, N1s, O1s, and C1s signals. Atomic force microscopy (AFM) analysis was performed for illustration 3D morphologies of biopolymer/silver nanocomposites and *Escherichia coli* features on a Bruker-Veeco-Digital Instruments Multi-Mode Atomic Force Microscope (Bruker, Germany). The dynamic mode (tapping) was applied during AFM imaging. For each sample, various regions were analyzed for the collection of representative results. The measurements were conducted using an antimony n-doped Si cantilever. For AFM imaging, the *Escherichia coli* strain control sample and a sample after the antibacterial test were collected, washed three times with phosphate-buffered saline, and centrifuged. The final sample was placed on an AFM microscope and measured by the tapping mode. The NanoScope Analysis software delivered from Bruker (Bruker, Germany) was applied for the data treatment. The surface morphology of the BP/AgP and BP/AgNP samples was studied by the field emission scanning electron microscopy (SEM) employing a Quanta^TM^ 3D FEG (FEI Company, Hillsboro, OR, USA) operating at 30 kV. The microstructure of the AgNP was analyzed by the transmission electron microscopy (TEM) Titan FEI Company operating at 300 kV. TEM analysis of the liquid phase with AgNP after ethanol extraction from the solid phase was performed. For this purpose, the drop of solution (AgNP extracted from the BP/AgP) onto a carbon-coated copper grid was placed, drying in the air, and measured at 300 kV.

Thermal analysis of BP, BP/AgP, and BP/AgNP samples was investigated using an STA 449 Jupiter F1 instrument, (Netzsch, Germany). During measurements, the samples in the form of clippings (10 mg) were heated in the atmosphere of synthetic air in the temperature range of 30–950 °C with a heating rate of 10 °C/min. The flow of synthetic air was stabilized on the level of 50 mL/min. Thermogravimetry (TG and DTG curves), differential scanning calorimetry (DSC), and FTIR spectra of the gaseous products were registered during analysis. The standard Al_2_O_3_ crucible and S type thermocouple were applied as a sensor of TG-DSC. An empty counterpart of the crucible was used as a reference in the same experimental cell. The gaseous products released during thermal decomposition were analyzed by coupled FTIR spectrometer (Bruker, Germany). The treatment of collected data was performed by the NETZSCH Proteus^®^ software (v6.1) delivered from the NETZSCH company (Netzsch, Germany).

## 4. Conclusions

In this work, nanomaterials based on ultra-small silver nanoparticles and the chitosan matrix as the nanofibers membrane was presented and investigated. In-situ reduction of the metallic phase can be a great idea for the synthesis and stabilization of the gentle nanophase, especially if ultra-small sizes of nanoparticles are required. The silver precursor in the form of diamminesilver(I) ions and their local stabilization by the biopolymer surface promotes the formation of the small size of silver nanoparticles and their homogeneous distribution in a solid membrane. The proposed approach confirms the presence of the natural reducing properties of chitosan directly related to the presence of -NH_2_ groups. The metallic nanoparticles were simply obtained by reduction, nucleation, and growth directly on the surface of the biopolymer material. The morphology of the Ag nanoparticles was spherical and the mean size of the nanoparticles was approximately 3 nm. These results are proof that the silver nanoparticles with ultra-small sizes are associated with the stabilization of the ionic form and may generate interaction with biopolymer functional groups and improve their stability on such binding arrangement. The X-ray photoelectron analysis of Ag3d photoelectrons and Auger analysis and Auger parameters as the sum of the binding energy of the Ag 3d5/2 photoelectron peak and kinetic energy of the Ag M_4_N_45_N_45_ and Ag M_5_N_45_N_45_ peaks were used to determine the chemical state of silver atoms on the biopolymer surface and confirm the stable ionic form of surface silver. Reliable information about their chemical composition and interaction with the support was confronted with the antibacterial properties of silver nanoparticles. It was found that the interaction effect depended on the size of the crystallites and might have a different effect on their interaction with the support surface and directly on their stability on the support body and finally with a possible interaction with the bacteria cell. The structural changes of the biopolymer after the silver modification was described by SAXS due to the SAXS effect generating by AgNP. Wide-angle X-ray diffraction allowed us to observe the differences in the size of the metallic nanophase and concisely determine the size of silver crystallites techniques in the first stage of the research. The visible broadening of the signals indicated a nano-dimensional effect. The improved thermal stability was confirmed for the BP/AgP sample. The improvement of thermal stability in the case of the/AgP material is related to the presence of a low-molecular silver phase on the surface of the biopolymer fibers, which creates a protective layer and improves the thermal properties of the carrier. Finally, the biostatic properties of the BP/AgP sample were reinforced compared to the BP/AgP sample. The impact of the proposed solution on this factor was discussed. The destruction of the bacteria cell after exposure to BP/AgP was visible on AFM images as significant morphological changes. Good correlation of visible damages with the proposed mechanism of the AgNP attack on *Escherichia coli* was achieved. The presented characteristics of investigated materials allow understanding of the mechanisms of the deposition of the metallic phase on the biopolymer surface, its stability including agglomeration capability, and assessment of potential applications. The nanofibers formed a three-dimensional network that would be suitable for membrane, filtration, and sorption applications.

## Figures and Tables

**Figure 1 ijms-21-09388-f001:**
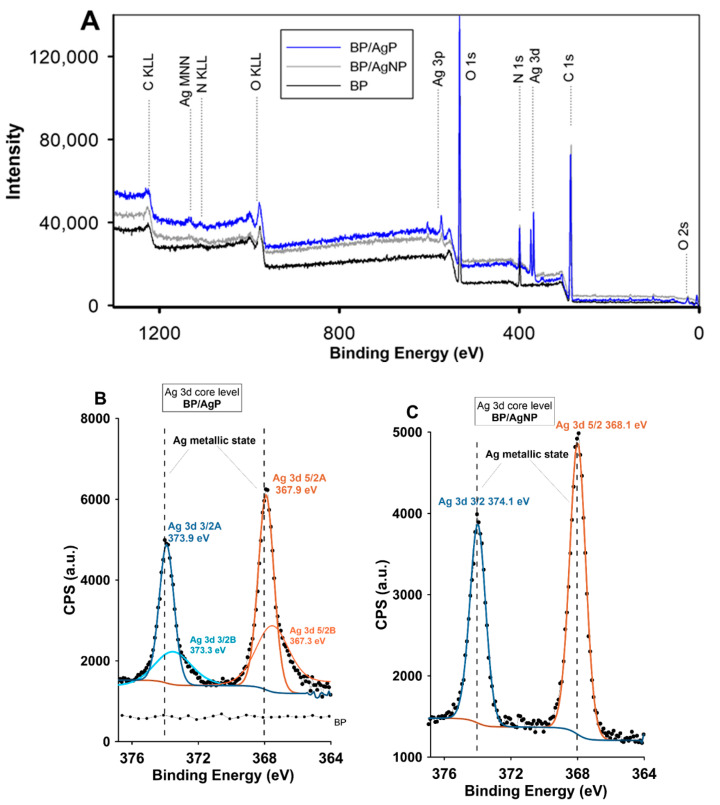
(**A**) XPS survey spectra for BP, BP/AgNP, and BP/AgP nanocomposites, (**B**) high–resolution XPS spectra of Ag3d core level of BP/AgP sample, and (**C**) high–resolution XPS spectra of the Ag3d core level of the BP/AgNP sample.

**Figure 2 ijms-21-09388-f002:**
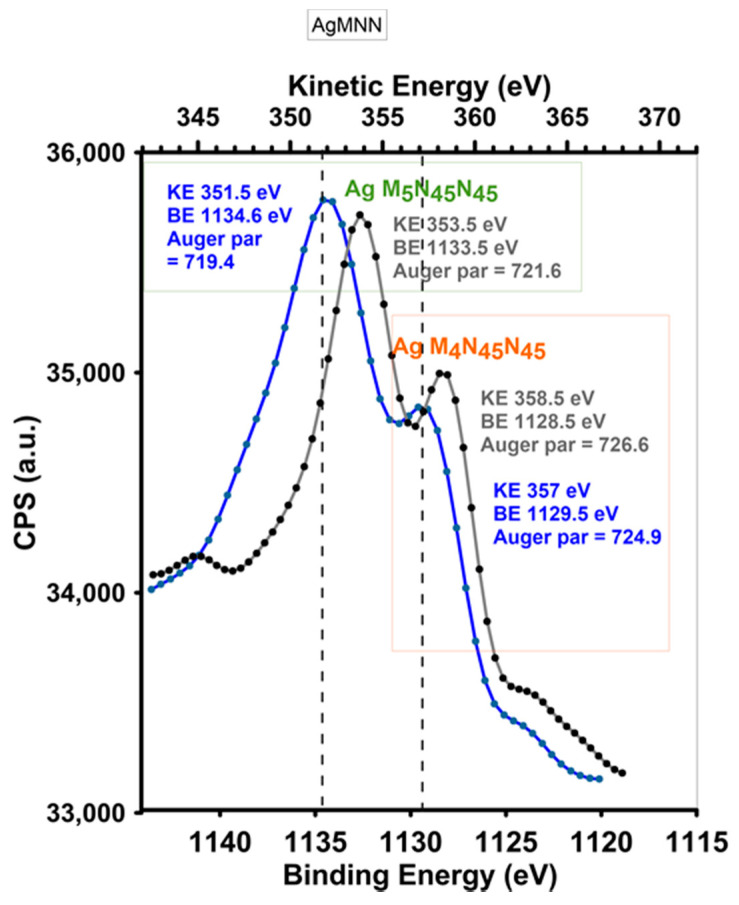
Ag MNN Auger electron spectra of BP/AgP (blue line) and BP/AgNP (grey line) materials. MNN implies the Auger emission type, MNN symbols in the transition label correspond to the three energy levels involved in the transition (in this case: M_5_N_45_N_45_ and M_4_N_45_N_45_).

**Figure 3 ijms-21-09388-f003:**
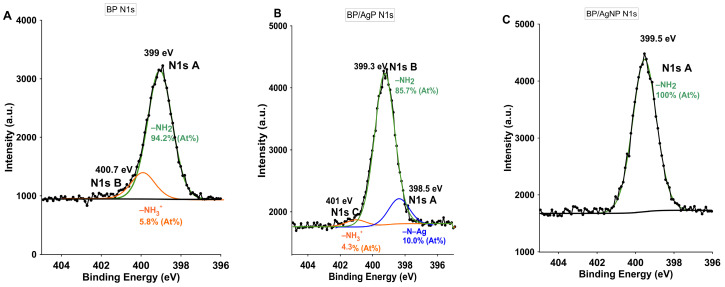
High-resolution XPS spectrum for the N1s core level for (**A**) BP, (**B**) BP/AgP, and (**C**) BP/AgNP.

**Figure 4 ijms-21-09388-f004:**
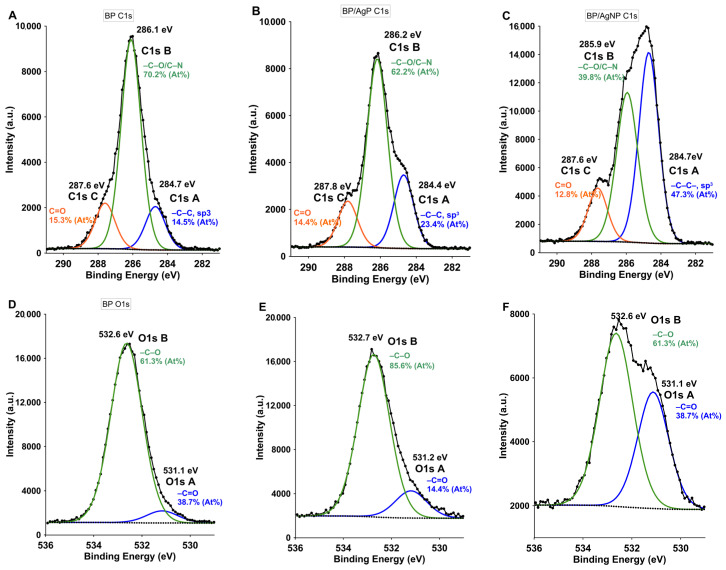
XPS high resolution spectra of C1s photoelectron lines for (**A**) BP, (**B**) BP/AgP, (**C**) BP/AgNP and O1s for (**D**) BP, (**E**) BP/AgP and (**F**) BP/AgNP samples.

**Figure 5 ijms-21-09388-f005:**
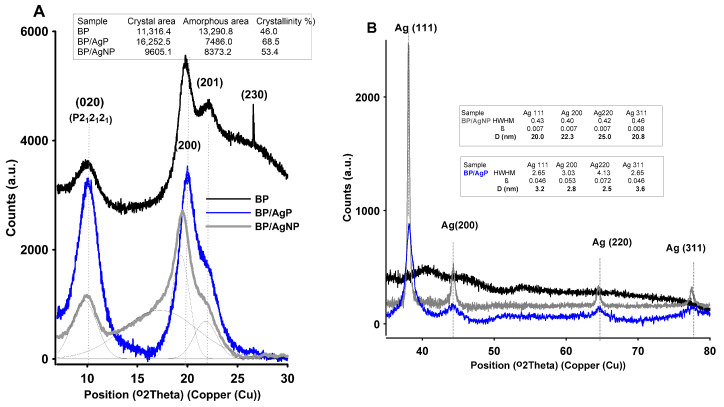
X-ray powder diffraction pattern of the investigated materials, BP, BP/AgP, and BP/AgNP. (**A**) XRD peaks between the range of 5–30°of 2θ of chitosan phase and (**B**) XRD peaks between the range of 30–80°of 2θ of crystal silver phase.

**Figure 6 ijms-21-09388-f006:**
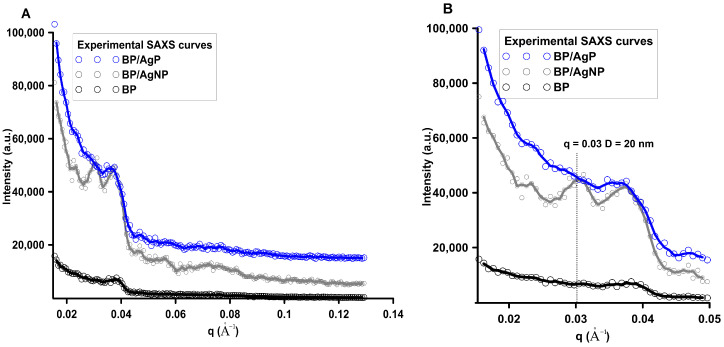
Experimental small-angle X-ray scattering (SAXS)profiles of BP, BP/AgP, and BP/AgNP samples in the linear (**A**) initial range of SAXS patterns in the logarithmic (**B**) scale.

**Figure 7 ijms-21-09388-f007:**
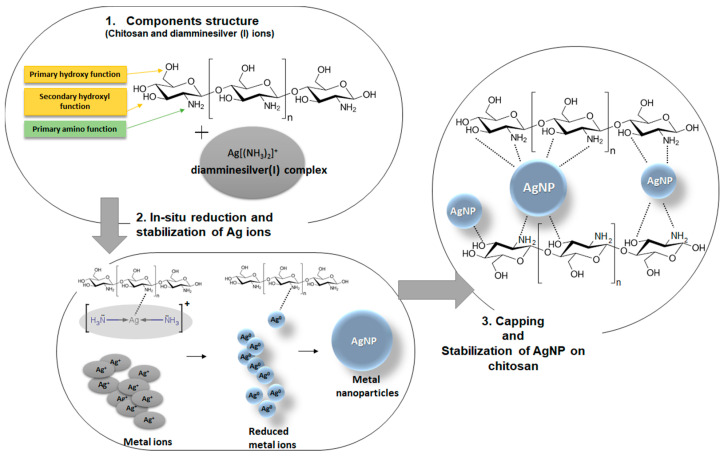
Schematic illustration of reduction and stabilization process of the diamminesilver(I) complex to AgNP by BP.

**Figure 8 ijms-21-09388-f008:**
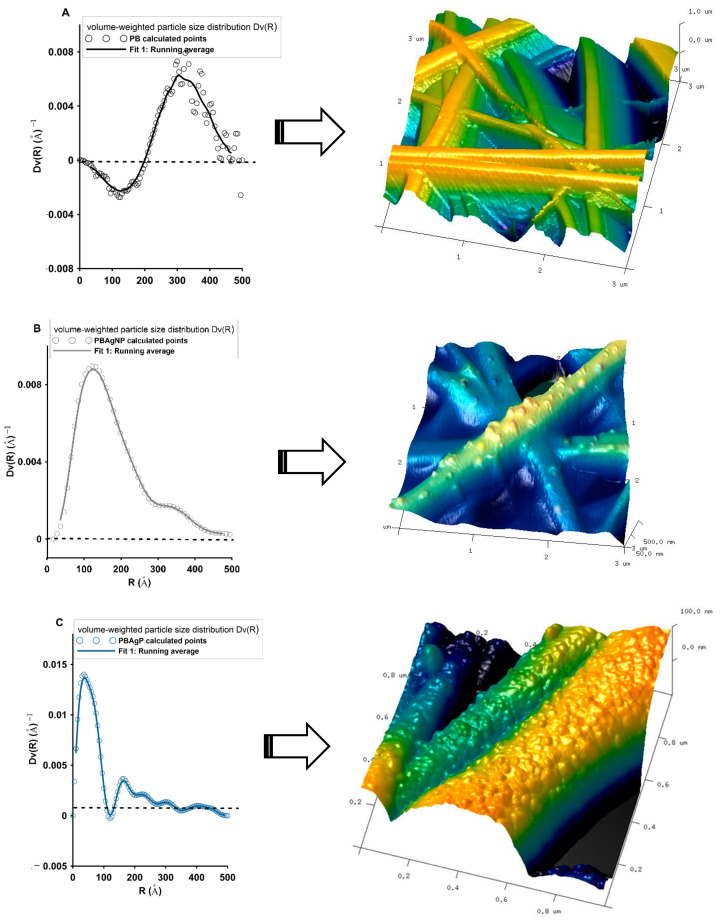
Volume-weighted particle size distribution Dv(R) from the scattering curves for investigated samples, (**A**) BP, (**B**) BP/AgNP, (**C**) BP/AgP, and AFM topography of the investigated materials.

**Figure 9 ijms-21-09388-f009:**
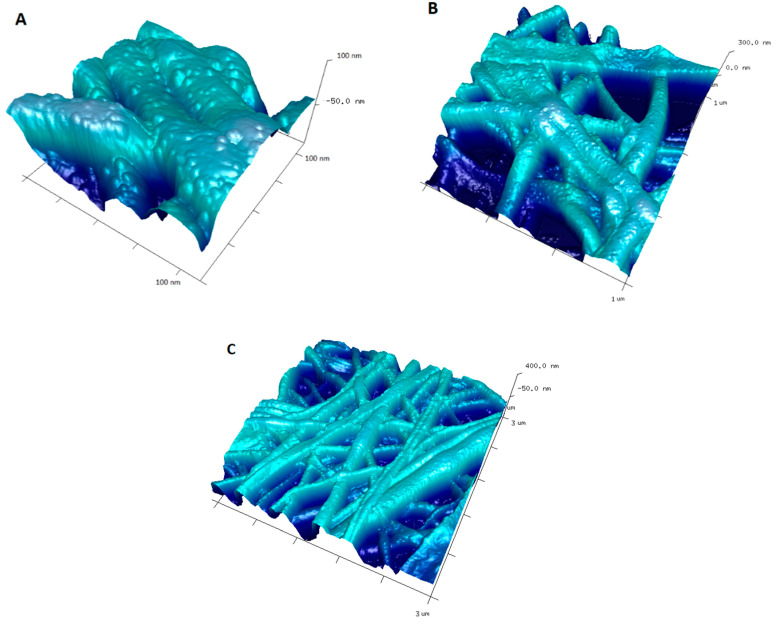
AFM 3D images representing the topography of BP/AgP nanocomposites at various magnifications, (**A**) zoomed area from 1 × 1 µm region, (**B**) full 1 × 1 µm region and (**C**) 3 × 3 µm region.

**Figure 10 ijms-21-09388-f010:**
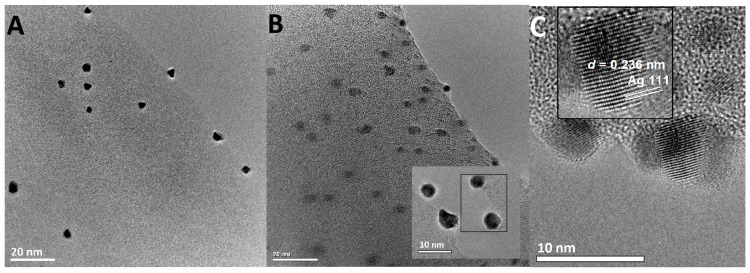
(**A**) TEM images of the AgNP extracted from BP/AgP sample as free AGNP. (**B**) TEM image of a small part of solid surface with AgNP (BP/AgP), scale bar = 20 nm. (**C**) the HRTEM image of selected AgNP with visible lattice planes and the interplanar spacing *d* = 0.236 nm corresponding to the Ag(111) lattice plane.

**Figure 11 ijms-21-09388-f011:**
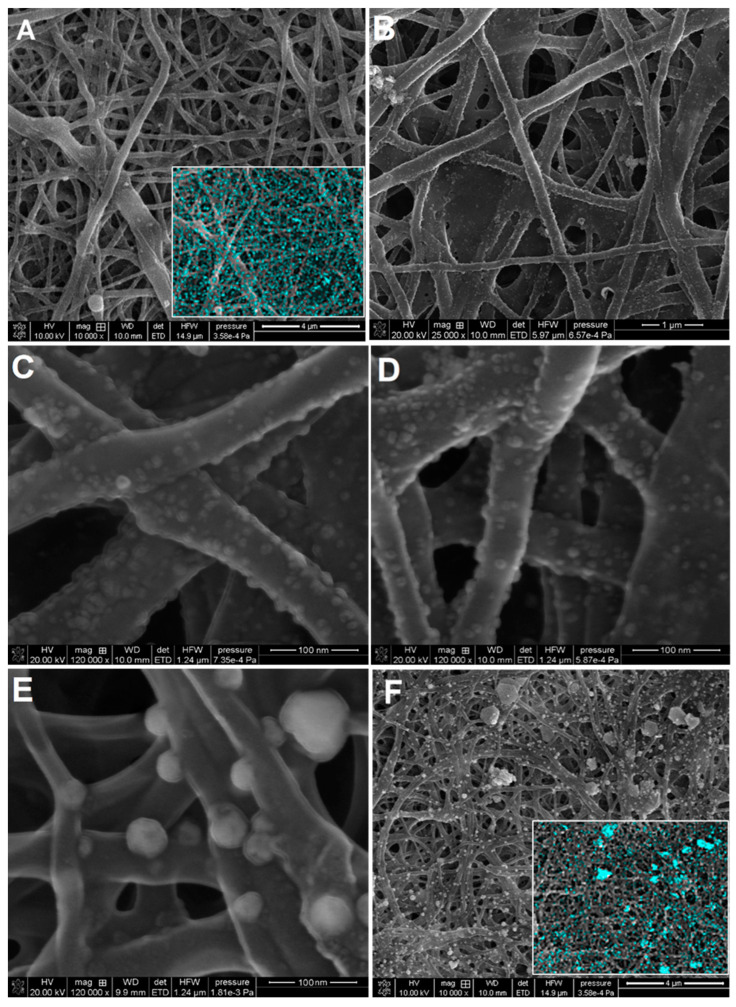
SEM images of the investigated nanocomposites, (**A**) BP, (**B**–**D**) BP/AgP, and (**E**,**F**) BP/AgNP. Insets presented in Figure 11A,F present the combination of SEM imaging with energy-dispersive X-ray spectroscopy (EDX). Blue objects represent the map of AgL signals.

**Figure 12 ijms-21-09388-f012:**
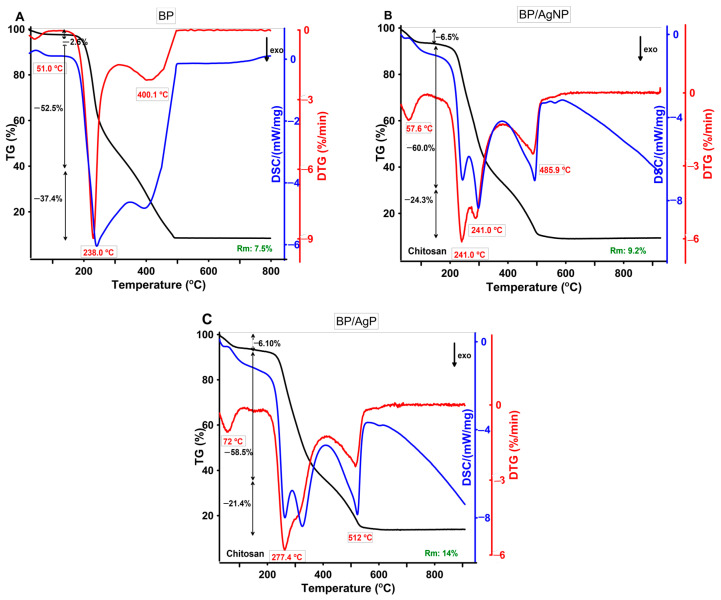
Thermogravimetry (TG), Differential thermogravimetry (DTG), and Differential Scanning Calorimetry (DSC) curves of BP (**A**), BP/AgNP (**B**) and BP/AgP (**C**) measured in the air atmosphere.

**Figure 13 ijms-21-09388-f013:**
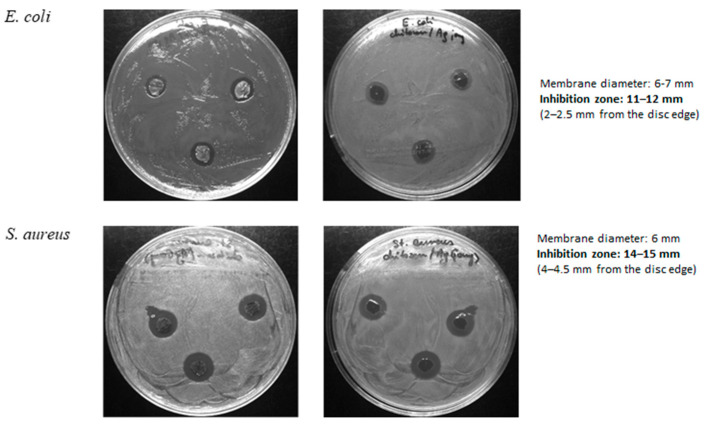
Images of the growth of *Escherichia coli* and *Staphylococcus aureus* bacteria strains on agar plates with BP/AgP.

**Figure 14 ijms-21-09388-f014:**
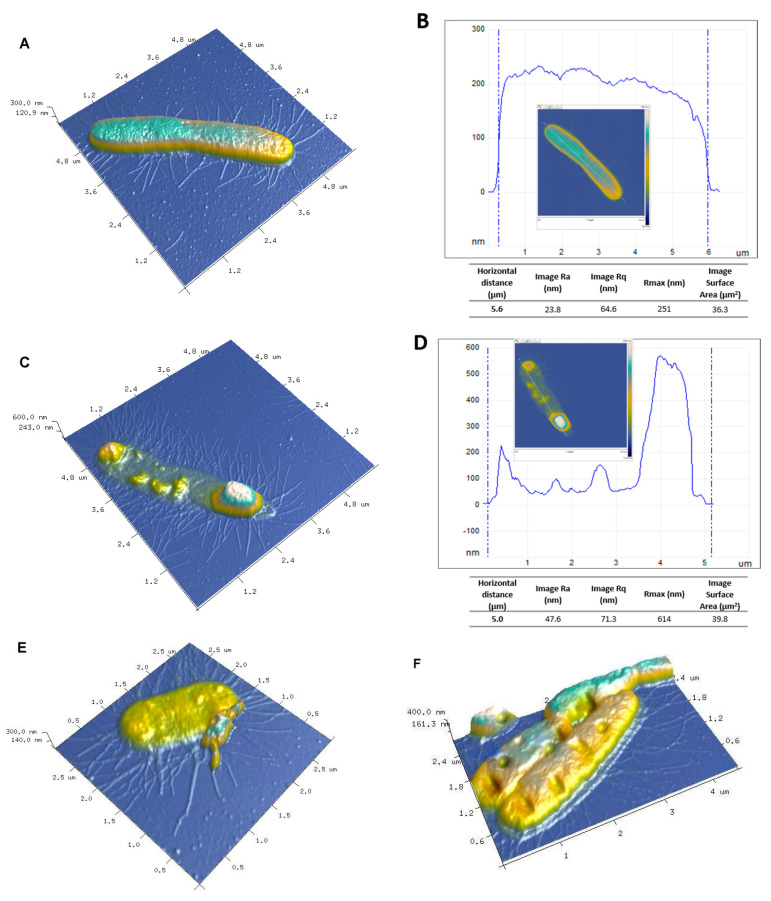
AFM images of *E. coli* untreated (control sample) and E. coli exposed to silver nanoparticles in BP/AgP, (**A**) topography as 3D view and (**B**) cross-section according to a selected line of the untreated sample, (**C**,**E**,**F**) AFM 3D visualization of the E. coli after exposure to BP/AgP and (**D**) cross-section according to a selected line of the treated bacteria presented in image (**C**) Tables include the roughness parameters for evaluating average surface roughness and morphological changes (Ra—roughness average defined as the arithmetic mean of the absolute values of the height of the surface profile, Rq—root mean square roughness, which is similar to the roughness average, however, include the mean squared absolute values of surface roughness profile; Rmax—the maximum profile peak height from the baseline, image surface area—a total area defined also for grains and masked areas).

**Figure 15 ijms-21-09388-f015:**
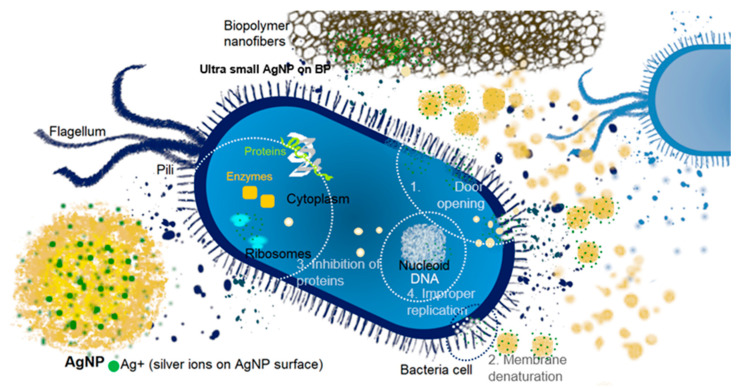
Scheme of the antibacterial actions of silver nanoparticles (AgNP) and silver ions (Ag+) against the bacteria model, illustrating the behavior of the systems considered in this work (AgNP in BP/AgNP (as big yellow AgNP) and BP/AgP (as small AgNP on biopolymer nanofibers).

**Figure 16 ijms-21-09388-f016:**
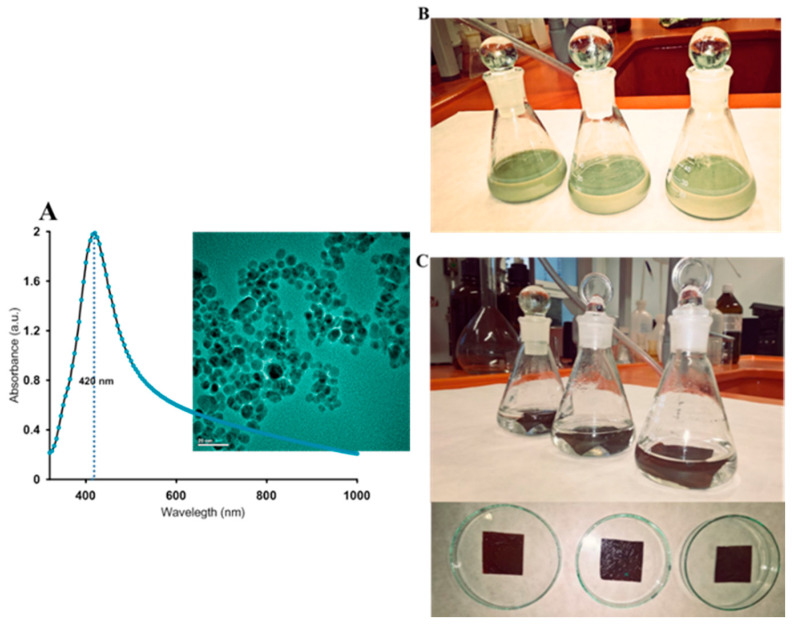
(**A**) UV–VIS absorption spectra of 20–30 nm silver nanoparticles solution (AgNP), (**B**) impregnation of the chitosan biopolymer by silver nanoparticles and preparation of the BP/AgNP sample, and (**C**) impregnation of the chitosan biopolymer by the silver precursor and preparation the BP/AgP sample.

**Table 1 ijms-21-09388-t001:** XPS results for BP, BP/AgP, and BP/AgNP samples. Positions of the main signal of photoelectrons from characteristic energetic levels.

Sample	Region	Position eV	Full Width at Half Maximum (FWHM)	Peak Area	Atomic Concentration (%)	Mass Concentration (%)	%SD
**BP**	C1s	286.1	2.47	200,979	62	55.1	0.1
	O1s	532.6	2.3	290,455	30.6	36.2	0.1
	N1s	399.1	2.27	38,384	6.6	6.8	0.1
**BP/AgP**	C 1s	286.1	2.58	201,199	59.1	45.7	0.2
	O 1s	533.1	2.52	298,570	29.9	30.9	0.1
	N 1s	399.6	2.27	41,630	6.8	6.1	0.2
	Ag 3d	368.1	2.06	108,161	2.7	13.3	0.1
**BP/AgNP**	C 1s	284.7	2.68	222,346	71.8	59.3	0.2
	O 1s	531.7	2.57	157,888	17.4	19.1	0.1
	N 1s	399.2	2.14	42,796	7.7	7.4	0.1
	Ag 3d	367.7	1.92	78,518	1.4	10.4	0.1

**Table 2 ijms-21-09388-t002:** Characteristics of XPS peaks for BP/AgP and BP/AgNP from high-resolution analysis and their assignment.

Sample	Region	Peaks Position (eV)	Full Width at Half Maximum (FWHM)	Line Shape	Area	Atomic Concentration (%)	Binding Assignation
**BP**	C 1s A	284.7	1.35	GL(30)	2801.8	14.5	C–C sp3
	C 1s B	286.1	1.35	GL(30)	13,542.2	70.2	C–O
	C 1s C	287.6	1.35	GL(30)	2941.3	15.3	C=O
	O 1s A	531.2	1.55	GL(30)	1847.7	6.3	C=O
	O 1s B	532.6	1.55	GL(30)	27,326.6	93.7	C–O, SiO_2_
	N 1s A	399.0	1.46	GL(30)	3493.1	94.2	–NH_2_
	N 1s B	400.7	1.46	GL(30)	215.9	5.8	–NH_3_^+^
**BP/AgP**	C 1s A	284.7	1.3	GL(30)	4422.2	23.4	C–C sp3
	C 1s B	286.2	1.35	GL(30)	11,766.7	62.2	C–O
	C 1s C	287.8	1.28	GL(30)	2713.8	14.4	C=O
	O 1s A	531.2	1.55	GL(30)	4159.9	14.4	C=O
	O 1s B	532.7	1.55	GL(30)	24,817.7	85.6	C–O
	N 1s A	398.5	1.49	GL(30)	411.9	10.0	–N–Ag
	N 1s B	399.3	1.48	GL(30)	3523.3	85.7	–NH_2_
	N 1s C	401	1.52	GL(30)	175.4	4.3	–NH_3_^+^
	Ag 3d 5/2A	367.9	1.12	GL(30)	4648.1	47.4	Ag (I), Ag
	Ag 3d 3/2A	373.9	1.05	GL(30)	3078.2	31.5	Ag (I), Ag
	Ag 3d 5/2B	367.3	1.61	GL(30)	1236.2	12.6	Ag (I)
	Ag 3d 3/2B	373.3	1.55	GL(30)	834.01	8.5	Ag (I)
**BP/AgNP**	C 1s A	284.7	1.35	GL(30)	19,726.7	47.3	C–C sp3
	C 1s B	285.9	1.45	GL(30)	16,612.1	39.8	C–O
	C 1s C	287.6	1.31	GL(30)	5352.9	12.8	C=O
	O 1s A	531.1	1.5	GL(30)	5932.2	38.7	C=O
	O 1s B	532.6	1.6	GL(30)	9407.1	61.3	C–O
	N 1s	399.5	1.46	GL(30)	4231.8	100	–NH_2_
	Ag 3d 5/2	368	1.13	GL(30)	4406.7	100	Ag
	Ag 3d 3/2	374	1.11	GL(30)	2952.5	–	Ag

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
