# Peer review of "Small AgNP in the Biopolymer Nanocomposite System"

_ijms, 2020, doi:10.3390/ijms21249388_

Round 1
Reviewer 1 Report
- The abstract should be further improved, it should be focused on the background, results and significance, not the methods.
- This manuscript are written to focused on the chemical properties of the BP/AgNPs system, but for my understanding, the BP/AgNPs system are used for biological applications, so more attentions should be paid on their potential biological uses in manuscript writing. Otherwise, this work is more suitable to submit for a material or chemical journal.
- As indicated in the first review comments, although the nanosystem is well characterized, more biological studies are needed for further consideration in IJMS. Authors provide their speculation about the anti-bacterial mechanisms of AgNPs based on some references, however, the BP/AgNPs system might show different inhibition effects or mechanism against bacteria. Authors should provide more biological experiment to prove these speculations. Mechanism studies must be done before next submission.
- Authors should also compare the inhibition effects of BP alone, AgNPs alone and BP/AgNPs system, unless the authors admit that the proposed BP/AgNPs system showed no advantages beyond AgNPs. Some discussions are also needed to explain why BP/AgNPs system are needed.
Author Response
List of changes in the revised manuscript and detailed responses to Reviewers
- The abstract should be further improved, it should be focused on the background, results, and significance, not the methods.
Thank you for your kind comment. The abstract was corrected according to suggestions.
“Abstract: In this work ultra-small and stable silver nanoparticles were prepared by in-situ reduction of the diamminesilver(I) complex ([Ag(NH3)2]+) as the silver precursor (AgP) by functional groups located on the surface of the chitosan biopolymer. The work aimed to create a functional membrane system with biological (biostatic) properties. The main points of this report refer to using the least necessary AgNP amount (environmentally friendly aspects fulfilled by reduction at the destination and avoiding their overproduction), usage of the natural functionality, and reducing properties of biopolymers (BP) without additional steps of their modification as well, confirmation of bioactive properties of new materials. These steps pave the way for generating the metallic phase directly on the biopolymer surface and producing ultra-small (~3 nm) metallic crystallites with a high degree of homogeneity which may be linked to their stabilization by the biopolymer surface. This provided an Ag concentration of 13.3 mass % and 2.7 atomic % for Ag-doped material (determined by XPS) and pointed at the high affinity of [Ag(NH3)2]+ ions to chitosan functional groups and high enterability of silver to the biopolymer surface. The small size of silver nanoparticles as an effective and stable source of silver ions, their crystal quality, and homogeneous distribution in a whole network of the solid membrane was confirmed by appropriate microscopic techniques. The results obtained concluded that the biopolymer membrane properties have been significantly improved by the integration with ultra-small Ag nanoparticles which adds value to its applications as a biostatic membrane system for filtration and separation issues. The nanometric surface-sensitive XPS analysis (the analysis depth is ~10 nm) was applied to obtained information about the elemental composition, chemical concentration, and chemical state of surface atoms. It was found that ultra-small metallic nanoparticles create a steady source of silver ions what is strictly related to enhanced their biostatic properties. Moreover, it was established that very small silver nanoparticles with disturbed electronic structure and plasmonic properties may generate interaction between amine groups from a biopolymer body which enables building an improved homogeneity of nanometallic layer on the biopolymer surface. Furthermore, not very developed research of Ag NPs by XPS and Auger analysis should be completed to establish a correlation between the chemical nature of the nanocomposite surface and the antibacterial or biostatic properties of proposed biomaterials. In this work, the significant differences between the typical way (deposition of ex-situ-prepared AgNP) and the proposed in-situ synthesis approach were determined. The improved thermal stability was confirmed for a sample with small AgNP (BP/AgP) and explained as the effect of the presence of the protective layer of a low-molecular silver phase. Finally, the antibacterial activity of BP/AgP nanocomposite was tested using Gram-positive (S. aureus), and the Gram-negative (E. coli) bacteria biofilms. The grafted membrane with AgNP has shown clear inhibition properties by destruction and multiple damages of bacteria cells. It was found that greater biocide activity of BP/AgP than BP/AgNP material should be associated with the size and stability of ultra-small AgNP. The possible mechanisms of biocidal activity were discussed, and the investigation of the AgNP influence on the bacteria body was illustrated by AFM measurements.”
- This manuscript are written to focused on the chemical properties of the BP/AgNPs system, but for my understanding, the BP/AgNPs system are used for biological applications, so more attentions should be paid on their potential biological uses in manuscript writing. Otherwise, this work is more suitable to submit for a material or chemical journal.
Thank you for this comment. The additional sentences were added to the manuscript:
Line 13 “The work aimed to create a functional membrane system with biological (biostatic) properties.”
Line 28 “The results obtained concluded that the biopolymer membrane properties have been significantly improved by the integration with ultra-small Ag nanoparticles which adds value to its applications as a biostatic membrane system for filtration and separation issues.”
Line 115 The system of silver nanoparticles deposited on a fibrous biopolymer substrate is important from the point of view of practical applications, especially biomedical ones. The most overriding elements include making the basis of wound repair systems with advanced and controlled drug release mechanisms [42-44]. When the AgNP are applied their stability and bonding strength may be responsible for various adhesion properties and provide a specified rate of Ag+ release. This in turn translates into antibacterial properties and determines the usefulness of such a system. Recent studies of similar multicomponent systems (chitosan/silver-NPs) systems suggest synergistic antibacterial effects achieved by combining chitosan with Ag-NPs which increases the importance of water filtration systems [45]. The stability of the incorporated AgNP phase allows building improved nanofiltration systems for purification of drinking water and other substances [46-48]. The idea may be e.g. a membrane system for the filtration of liquid systems (e.g. milk) with the simultaneous sorption of appropriate biomolecules e.g. cholesterol [49-52].
Additional References were added to the reference list:
[42] Zhou, W.; Li, Y.; Yan, J.; Xiong, P.; Li, Q.; Cheng, Y.; Zheng, Y. Construction of Self-defensive Antibacterial and Osteogenic AgNPs/Gentamicin Coatings with Chitosan as Nanovalves for Controlled release. Scientific Reports 2018, 8, 13432.
[43] Guo, R.; Wen, J.; Gao, Y.; Li, T.; Yan, H.; Wang, H.; Niu, B.; Jiang, K. Effect of the adhesion of Ag coatings on the effectiveness and durability of antibacterial properties. Journal of Materials Science 2018, 53, 4759-4767.
[44] Abdelgawad, A.; Hudson, S.; Rojas, O. Antimicrobial wound dressing nanofiber mats from multicomponent (chitosan/silver-NPs/polyvinyl alcohol) systems. Carbohydrate polymers 2014, 100, 166-178.
[45] Adibzadeh, S.; Bazgir, S.; Katbab, A. A. Fabrication and characterization of chitosan/poly(vinyl alcohol) electrospun nanofibrous membranes containing silver nanoparticles for antibacterial water filtration. Iranian Polymer Journal 2014, 23, 645-654.
[46] Adibzadeh, S.; Bazgir, S.; Katbab, A. A. Fabrication and characterization of chitosan/poly(vinyl alcohol) electrospun nanofibrous membranes containing silver nanoparticles for antibacterial water filtration. Iranian Polymer Journal 2014, 23, 645-654.
[47] Wang, L.; Zhang, C.; Gao, F.; Pan, G. Needleless electrospinning for scaled-up production of ultrafine chitosan hybrid nanofibers used for air filtration. RSC Advances 2016, 6, 105988-105995.
[48] Grimmelsmann, N.; Homburg, S. V.; Ehrmann, A. Electrospinning chitosan blends for nonwovens with morphologies between nanofiber mat and membrane. IOP Conference Series: Materials Science and Engineering 2017, 213, 012007.
[49] Bokura, H.; Kobayashi, S. Chitosan decreases total cholesterol in women: a randomized, double-blind, placebo-controlled trial. Eur J Clin Nutr 2003, 57, 721-725.
[50] Park, J. H.; Hong, E.-K.; Ahn, J.; Kwak, H.-S. Properties of nanopowdered chitosan and its cholesterol lowering effect in rats. Food Science and Biotechnology 2010, 19, 1457-1462.
[51] Tao, Y.; Zhang, H.; Gao, B.; Guo, J.; Hu, Y.; Su, Z. Water-Soluble Chitosan Nanoparticles Inhibit Hypercholesterolemia Induced by Feeding a High-Fat Diet in Male Sprague-Dawley Rats. Journal of Nanomaterials 2011, 2011, 814606.
[52] Ylitalo, R.; Lehtinen, S.; Wuolijoki, E.; Ylitalo, P.; Lehtimäki, T. Cholesterol-lowering properties and safety of chitosan. Arzneimittelforschung 2002, 52, 1-7.
- As indicated in the first review comments, although the nanosystem is well characterized, more biological studies are needed for further consideration in IJMS. Authors provide their speculation about the anti-bacterial mechanisms of AgNPs based on some references, however, the BP/AgNPs system might show different inhibition effects or mechanisms against bacteria. Authors should provide more biological experiment to prove these speculations. Mechanism studies must be done before next submission.
Additional biological studies include the atomic force microscopy (AFM) of bacteria morphology. Especially, the investigation of the changes in biofilm morphology before and after exposure to AgNP in BP/AgNP was investigated. The AFM images of E. Coli before and after contacting with antibacterial material (BP/AgP) were presented as 3D surface visualization. Unfortunately, we have no other data in this regard. AFM analysis was performed for control purposes only for the E. Coli system. However, the presented results, comply with the indicated anti-bacterial mechanisms and significant changes in cell morphology indicate the correctness of presented theoretical assumptions.
Page 21: The additional figure (Figure 14) was added to the manuscript.
The additional results with their discussion were added to the manuscript:
Line 743 “Atomic force microscopy (AFM) can be used to evaluate the surface morphology of a biological specimen changed as a result of interaction with a harmful factor. Fig. 14 A shows an AFM image of E. coli control cells. The cell exhibit a characteristic rod shape with a size above 5 µm. The E. coli cells before treatment were characterized by a homogeneous surface without visible and significant damage. Fig.14C-14F illustrates E. coli bacteria after exposure to BP/AgP nanocomposite. After contact of E.coli with BP/AgP, some features of bacteria destruction were defined. Mild and extensive membrane degradation effects, additional debris, and small fragments located on the bacteria surface and their vicinity as well as atypically local accumulation of cellular fragments were observed (Fig. 14C). Moreover, significant cellular collapse, lysis of the cell contents (Fig. 14E) as well as holes, and shortages of biological material were visible (Fig. 14F). The cross-section profiles were created by selecting one line along with the bacteria cell of a two-dimensional AFM scan (presented as an inset in Fig. B and D) and subsequently plotting as distance vs. height data. The roughness of the bacteria body becomes different after AgNP treatment. Here the Ra, Rq, Rmaxwere higher for damaged bacteria and clearly illustrated the structure and nanoarchitecture features as the reflection of the interactions of a material with its external environment. Furthermore, cross-section line profiles were generated for untreated bacteria and bacteria after contact with BP/AgNP surface and presented in Fig. B and D, respectively The cross-section of bacteria without damage was smooth and free of unexpected surface irregularities, while such features were observed for the sample after exposure to AgNP. The set of morphologies observed indicates cell damage and confirm the presence of steps mechanism to the cytotoxic effect of small AgNP against E. coli as an example of real microbial biosystems. The explanation of the visible changes is discussed below and illustrated in Fig 15.”
“Figure. 14. AFM images of E. coli untreated (control sample) and E.coli exposed to silver nanoparticles in BP/AgP, (A) topography as 3D view and (B) cross-section according to a selected line of the untreated sample, (C, E, F) 3D visualization of the E. coli after exposure to BP/AgP. Tables include the roughness parameters for evaluating average surface roughness and morphological changes (Ra – Roughness average defined as the arithmetic mean of the absolute values of the height of the surface profile, Rq - Root mean square roughness which is similar to the roughness average, however, include the mean squared absolute values of surface roughness profile; Rmax -The maximum profile peak height from the baseline, Image Surface Area - a total area defined also for grains and masked areas).”
- Authors should also compare the inhibition effects of BP alone, AgNPs alone, and BP/AgNPs system unless the authors admit that the proposed BP/AgNPs system showed no advantages beyond AgNPs. Some discussions are also needed to explain why BP/AgNPs system is needed.
The inhibition effect of BP and AgNP was discussed in our previous paper. To avoid the unnecessary repetition of these data their correlation was advisable and the suitable reference was incorporated in the manuscript:
Line 720 “The results show that the small size of the nanoparticles and their homogeneous distribution on the support surface cause greater biocide activity than BP/AgNP material with the greater size of AgNP. Thus the effect of increasing the bioactive properties with decreasing size was confirmed by increasing the inhibition zone from 10mm and 13 mm (for BP/AgNP) to 11-12mm and 14-15mm (for BP/AgP) for E. coli and S. aureus respectively. Moreover, neither strain showed a significant zone of inhibited growth around BP alone discs. Furthermore, it was found that AgNP suspension inhibited the growth of bacteria in a dose-dependent manner. When the same amount of AgNP (1 µl of AgNP dropped on the composite discs and 1 µl used directly) was used, the better activity of AgNP stabilized on the biopolymer was observed (the synergy between biopolymer and silver nanoparticles). The detailed biostatic activity of BP alone, AgNPs alone, and BP/AgNPs are presented in our previous work [41] and should be compared with a significantly more active BP/AgP system. “

Reviewer 2 Report
Zienkiewicz-Strzałka & Deryło-Marczewska report the synthesis of a biopolymeric system containing silver nanoparticles with dimensions below 10 nm generated from in-situ reduction of silver ions using as a precursor diamminesilver(I) complex solution. The advantages of their approach were the miniaturization of metallic particles with average size of 3 nm, the higher homogeneity and the improved stability compared to those produced by deposition of ex-situ-prepared AgNP. Overall, the study is valuable and the advantages of the proposed approach were confirmed by the obtained results. The authors are presenting new results, the article is well written, well presented and discussed. I would recommend the publication after minor revision.
Minor points
- The sentence “The general assumptions of the work apply the excess production of the nanometer metallic phase” is not clear, please revise it.
- Line 19 (abstract) : It is not clear what the authors mean with the term “applying the effect of local stabilization”
- Line 23 (Abstract) : I believe that instead of “nanometers” the term nanometric would be more appropriate?
- Line 27(Abstract): proposed biomaterials instead of “ biomaterials proposals”
- Notations for figure 12 must be corrected, instead of “PB”, BP as used by authors throughout the manuscript.
Author Response
List of changes in the revised manuscript and detailed responses to Reviewers
- The sentence “The general assumptions of the work apply the excess production of the nanometer metallic phase” is not clear, please revise it.
Thank you for your kind comment. The sentence was corrected.
Line 15 “The main points of this report refer to used least necessary AgNP amount (environmentally friendly aspects fulfilled by reduction at the destination and avoids their overproduction”.
- Line 19 (abstract): It is not clear what the authors mean with the term “applying the effect of local stabilization”
Thank you for your comment. The sentence was corrected.
Line 22 “which may be linked to their stabilization by the biopolymer surface.”
- Line 23 (Abstract): I believe that instead of “nanometers” the term nanometric would be more appropriate?
Thank you for your comment. It was corrected (Line 31).
- Line 27(Abstract): proposed biomaterials instead of “ biomaterials proposals”
Thank you for your comment. It was corrected (Line 40).
- Notations for figure 12 must be corrected, instead of “PB”, BP as used by authors throughout the manuscript.
Thank you for your comment. It was corrected (Page 18).
Other minor corrections were marked as yellow points in the manuscript.
Round 2
Reviewer 1 Report
Manucript are well revised, but the abstract might need to be further simplified.
Author Response
The abstract part was simplified and condensed.
Abstract: In this work, ultra-small and stable silver nanoparticles on chitosan biopolymer (BP/AgP) were prepared by in-situ reduction of the diamminesilver(I) complex ([Ag(NH3)2]+) to create a biostatic membrane system. The small AgNP (~3 nm) as a stable source of silver ions, their crystal form, and homogeneous distribution in the whole solid membrane were confirmed by TEM, SEM, and AFM. The XPS and Auger analysis were applied to investigate the elemental composition, concentration, and chemical state of surface atoms. It was found that ultra-small metallic nanoparticles may form a steady source of silver ions and enhance the biostatic properties of solid membranes. Ultra-small AgNP with disturbed electronic structure and plasmonic properties may generate interaction between amine groups of biopolymer for improving the homogeneity of nanometallic layer. In this work, the significant differences between the typical way (deposition of ex-situ-prepared AgNP) and the proposed in-situ synthesis approach were determined. The improved thermal stability (TG/DSC analysis) for BP/AgP was observed and explained by the presence of the protective layer of a low-molecular silver phase. Finally, the antibacterial activity of BP/AgP nanocomposite was tested using selected bacteria biofilms. The grafted membrane showed clear inhibition properties by destruction and multiple damages of bacteria cells. The possible mechanisms of biocidal activity were discussed, and the investigation of the AgNP influence on the bacteria body was illustrated by AFM measurements. The results obtained concluded that the biopolymer membrane properties have been significantly improved by the integration with ultra-small Ag nanoparticles which adds value to its applications as a biostatic membrane system for filtration and separation issues.
This manuscript is a resubmission of an earlier submission. The following is a list of the peer review reports and author responses from that submission.
Round 1
Reviewer 1 Report
This paper is very similar to their previous paper (Ref. 49; Zienkiewicz-Strzałka, M. D.-M., A.; Skorik, Y.A.; Petrova, V.A.; Choma, A.; Komaniecka, I. Silver Nanoparticles on Chitosan/Silica Nanofibers: Characterization and Antibacterial Activity. Int J Mol Sci 2019, 21, 166-186.). So, the authors should estimate the more significant effect of BP/AgP than those of Ag/CS/silica nanofibers. Therefore, this paper should not be accepted without adding the novelty of data of BP/AgP.
Reviewer 2 Report
This work reports the synthesis of ultra-small and stable silver nanoparticles in a biopolymer carrier with a study of their comparative antibacterial activity. Although the nanosystem is well characterized, more biological studies are needed for further consideration in JJMS.
- Authors should provide more comments about the advantages of the proposed small AgNP in the biopolymer nanocomposite beyond other reported biopolymer-AgNP system.
- Abstract should be simplified and re-designed.
- TEM images of the Ag NPs should also be provided.
- English writing should be further improved, for example: “the desired amount of diamminesilver(I) solution was this solution”, and “The silver precursor in the form of diamminesilver(I) ions and their local stabilization by the biopolymer surface determines the small 440 size of silver nanoparticles. ”.
- Mechanism studies about the anti-bacterial effects of biopolymer-AgNP system should be provided as IJMS is a journal focusing on molecular science.